# GRAM-DTI: Adaptive Multimodal Representation Learning for Drug–Target Interaction Prediction

**Feng Jiang**[1*], **Amina Mollaysa**[2*], **Hehuan Ma**[1], **Yuzhi Guo**[1], **Tommaso Mansi**[2],
**Junzhou Huang**[1], **Mangal Prakash**[2†], **Rui Liao**[2] [†]
[1]University of Texas at Arlington, [2]Johnson & Johnson Innovative Medicine

## Abstract

Drug target interaction (DTI) prediction is a cornerstone of computational drug discovery, enabling rational design, repurposing, and mechanistic insights. While deep learning has advanced DTI modeling, existing approaches primarily rely on SMILES–protein pairs and fail to exploit the rich multimodal information available for small molecules and proteins. We introduce GRAM-DTI, a pretraining framework that integrates multimodal molecular and protein inputs into unified representations. GRAM-DTI extends volume-based contrastive learning to four modalities, capturing higher-order semantic alignment beyond conventional pairwise approaches. To handle modality informativeness, we propose adaptive modality dropout, dynamically regulating each modality's contribution during pre-training. Additionally, IC50 activity measurements, when available, are incorporated as weak supervision to ground representations in biologically meaningful interaction strengths. Experiments on four publicly available datasets demonstrate that GRAM-DTI consistently outperforms state-of-the-art baselines. Our results highlight the benefits of higher-order multimodal alignment, adaptive modality utilization, and auxiliary supervision for robust and generalizable DTI prediction. Our code is available at `https://github.com/uta-smile/GRAM-DTI`.

## 1 Introduction

Drug target interaction (DTI) prediction is a central challenge in computational drug discovery, underpinning applications in rational drug design, repurposing of approved drugs, and elucidation of mechanisms of action (Vefghi et al., 2025; Jiang et al., 2024b; Ma et al., 2022). Traditional experimental screening, though reliable, is prohibitively expensive and cannot feasibly cover the vast chemical and proteomic search space (Jiang et al., 2024a; Ma et al., 2024). Computational methods therefore play an increasingly critical role in prioritizing candidate drug–protein pairs for experimental validation, accelerating discovery pipelines and reducing cost (Panahandeh & Mansouri, 2025; Liao et al., 2025).

DTI prediction methods have evolved from similarity-based and network-based heuristics to machine learning and, more recently, deep learning approaches (Shi et al., 2024; Panahandeh & Mansouri, 2025). Early methods relied on molecular similarity or interaction propagation but struggled with generalization. Modern neural models, including graph neural networks and sequence-based architectures now dominate, learning directly from raw SMILES and amino acid sequences (Peng et al., 2024; Zhao et al., 2025; Liu et al., 2025; Xia et al., 2023). However, these approaches remain largely restricted to SMILES–protein pairs, overlooking the richer multimodal information available for molecules and proteins that could yield more robust and generalizable interaction predictions.

While multimodal pre-training has been recently explored by a few works for DTI prediction (Lu et al., 2025; Ye et al., 2021; Chen et al., 2020), existing approaches suffer from three limitations. Firstly, they rely on pairwise contrastive learning anchored to a single modality. Such schemes cannot capture higher-order interdependencies as the number of modalities increases (Cicchetti et al.,

---

[*]These authors contributed equally as first authors. Correspondence to: `fxj8843@mavs.uta.edu`
[†]These authors contributed equally as senior (last) authors.

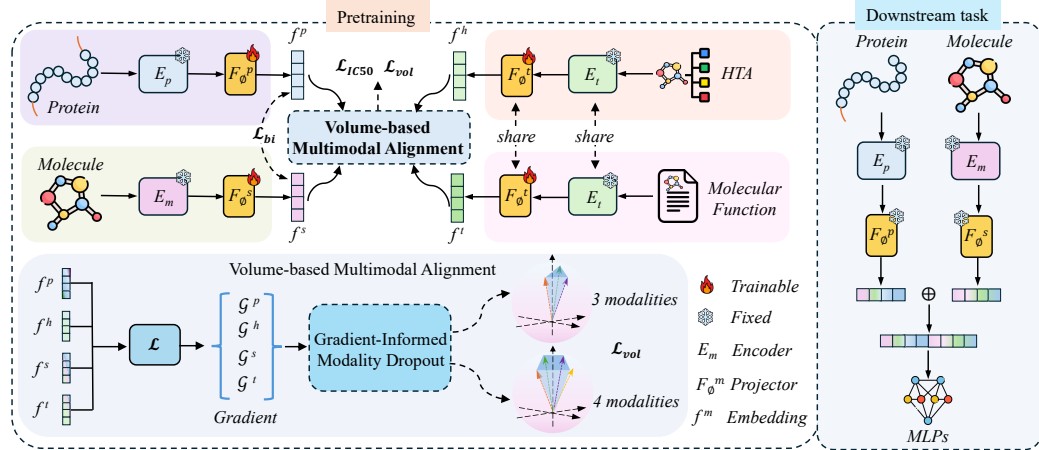

Figure 1: Overview of GRAM-DTI architecture. Left: pretraining phase with volume-based multimodal alignment across four modalities (SMILES, text, HTA, protein sequences). The framework uses gradient-informed adaptive modality selection to dynamically regulate modality contributions during training. Right: downstream task prediction.

2024). Secondly, they assume all modalities are equally informative, ignoring that data sources often differ in quality, completeness, and relevance across samples and training stages. Static fusion can therefore lead to suboptimal representations when dominant but less informative modalities overshadow complementary signals. Finally, valuable supervision signals such as IC50 activity measurements are publicly available for a subset of drug–protein pairs, yet they remain unutilized during pre-training despite their direct biological relevance for DTI prediction task.

To address these gaps, we propose GRAM-DTI, a novel multimodal pre-training framework specifically tailored for downstream DTI prediction task (see Fig. 1). To this end, we curate a high quality multimodal dataset consisting of diverse protein and small molecule modalities and adapt the recent volume-based contrastive learning strategies from other domains (Cicchetti et al., 2024; Jiang et al., 2025) for geometric alignment of these modalities. Unlike traditional contrastive learning techniques, this offers a theoretically principled and scalable approach for aligning multiple modalities. Beyond volume based contrastive learning, our framework is novel in its flexibility to learn to dynamically weight each modality based on its informativeness during pre-training while also supporting activity-based labels as auxiliary supervisory signals, when available. Our main contributions are as follows.

- We introduce GRAM-DTI, a pre-training framework for DTI that integrates multimodal small molecule protein and modalities into a unified representation with volume-based contrastive learning.

- We introduce adaptive modality dropout, dynamically regulating modality contributions during pre-training to prevent dominant but less informative modalities from overwhelming complementary signals.

- We leverage IC50 activity measurements as additional weak auxiliary supervision, grounding learned representations in biologically meaningful drug–target interactions.

- We demonstrate state-of-the-art performance across four public datasets and multiple evaluation settings relevant for real-world drug discovery applications.

## 2 RELATED WORKS

**Multimodal Molecular Representation Learning** Recent advancements in molecular representation learning have shifted towards integrating multiple data modalities to enhance predictive performance. For instance, frameworks like TRIDENT (Jiang et al., 2025) combine SMILES strings, hierarchical taxonomic annotations, and functional text of small molecules to capture richer molecular semantics. These approaches leverage contrastive learning pretraining to align diverse data

sources, which improves generalization across various molecular downstream tasks even in the absence of fully paired datasets. Beyond TRIDENT, several molecular foundation models have been introduced, including MolFM (Luo et al., 2023) and MolCA (Liu et al., 2023), which integrate molecular graphs, textual descriptions, and domain-specific annotations into unified representations. These works highlight the broader trend of leveraging multimodal pre-training to construct general-purpose molecular representations.

**Drug–Target Interaction (DTI) Prediction**   DTI prediction has traditionally relied on unimodal representations, such as SMILES strings for drugs and amino acid sequences for proteins. Early deep learning models such as DeepDTA (Öztürk et al., 2018), MT-DTI (Shin et al., 2019), and TransformerCPI (Chen et al., 2020) demonstrated the effectiveness of sequence-based architectures for interaction prediction. Beyond sequence-based methods, more recent work has explored graph neural networks and SE(3)-equivariant geometric deep learning models, such as GraphDTA (Nguyen et al., 2021) and EquiBind (Stärk et al., 2022), which leverage spatial and structural information of drugs and proteins to enhance binding affinity prediction. In parallel, knowledge graph–based methods such as NeoDTI (Wan et al., 2019) and Hetionet-based repurposing frameworks (Himmelstein et al., 2017) exploit biomedical networks to capture higher-order relations among drugs, targets, and diseases. More recently, multimodal approaches have been proposed to better capture the complexity of drug–target interactions. For example, MDTips (Xia et al., 2023) integrates knowledge graphs, gene expression profiles, and structural information, while MGNDTI (Peng et al., 2024) employs a multimodal graph neural network to improve robustness and generalization. Another emerging direction is pre-training with large-scale unlabeled data to mitigate the scarcity of labeled DTI pairs. For instance, DTIAM (Lu et al., 2025) introduces separate pretraining for drug and target modalities before merging the learned representations for DTI prediction.

**Modality Dropout**   Modality dropout techniques have been proposed to enhance the robustness of multimodal models by preventing over-reliance on any single modality. For instance, the Learnable Irrelevant Modality Dropout (IMD) method (Alfasly et al., 2022) selectively drops irrelevant modalities during training, improving performance in multimodal action recognition tasks. Additionally, approaches like aggressive modality dropout have been shown to mitigate negative co-learning effects and enhance model accuracy in multimodal settings (Magal et al., 2025). Beyond dropout, adaptive fusion mechanisms have also been investigated. Cross-attention and gating strategies (Tsai et al., 2019; Peng et al., 2024; Mollaysa et al., 2025) dynamically regulate modality contributions, while tensor fusion methods (Zadeh et al., 2017) capture higher-order interactions across modalities. These ideas inform the design of adaptive strategies in molecular contexts, where modality informativeness often varies across data sources and training stages.

Unlike existing works, our GRAM-DTI framework captures higher-order semantic relationships beyond simple pairwise alignment/fusion. Furthermore, to the best of our knowledge, we are the first to explore strategies for adaptive modality dropout in the context of DTI prediction.

## 3   METHODOLOGY

Building upon recent advances (Cicchetti et al., 2024; Jiang et al., 2025) in volume-based modality alignment for effective representation learning, we extend the foundational concept of volume loss (Cicchetti et al., 2024), originally formulated for audio-video-text data, to the domain of protein-small molecule interactions. We aim to learn a unified embedding space that: 1) captures semantic relationships across modalities; 2) remains robust when modalities vary in informativeness; and 3) improves downstream DTI prediction task.

Formally, assume a pretraining dataset $D = \{(x_i^s, x_i^t, x_i^h, x_i^p, \delta_{y_i}^{IC50})\}_{i=1}^N$, where $x_i^s$, $x_i^t$, $x_i^h$, and $x_i^p$ denote the SMILES sequence, textual description of molecule, hierarchical taxonomic annotation (HTA) (Jiang et al., 2025) of molecule, and protein sequence, respectively. The variable $\delta_{y_i}^{IC50}$ indicates the IC50 activity class $y_i^{IC50}$ if a measured IC50 value is available for the protein–molecule pair $(x_i^p, x_i^s)$, and 0 otherwise. As illustrated in Fig. 1, we employ pre-trained encoders $E_i$ (Mol-Former (Ross et al., 2022) for SMILES, MolT5 (Edwards et al., 2022) for text and HTA, and ESM-2 (Lin et al., 2023) for proteins) to obtain initial modality-specific embeddings.

To keep pre-training efficient and scalable, we freeze the backbone encoders and train lightweight neural projectors $F_\phi^m$ that map each modality embedding into a shared representation space where they are semantically aligned. The resulting projected embeddings are denoted $f^m$, where $m \in \{SMILES, text, HTA, protein\}$.

## 3.1 GRAMIAN VOLUME-BASED MULTIMODAL ALIGNMENT

In contrast to traditional multimodal representation learning approaches which have been known to fail in capturing the complex interdependencies among three or more modalities (Cicchetti et al., 2024; Jiang et al., 2025), volume loss uses Gramian volume-based alignment of modalities ensuring semantic coherence across all modalities simultaneously.

**Gramian Volume**   Given embeddings $f_i^s, f_i^t, f_i^h, f_i^p \in \mathbb{R}^d$ that are learned from the four modalities $x_i^s, x_i^t, x_i^h, x_i^p$ respectively, we first normalize them such that $\|f_i^m\|_2 = 1$. We can then construct the Gram matrix $G \in \mathbb{R}^{4 \times 4}$ where

$$G_{kj} = \langle f_i^k, f_i^j \rangle, \ \ k, j \in \{s, t, h, p\} \tag{1}$$

The 4-dimensional volume spanned by these embedded vectors is equal to the square root of the determinant of the Gramian matrix (Cicchetti et al., 2024): $V(f_i^s, f_i^t, f_i^h, f_i^p) = \sqrt{\det(G)}$. From multimodal alignment perspective, smaller volume intuitively suggests stronger semantic alignment, as the embeddings occupy a more compact and cohesive subspace and vice-versa.

**Volume-Based Contrastive Loss**   Given the Gramian volume, contrastive objective is cast as volume minimization/maximization. As proposed in (Cicchetti et al., 2024), to construct negative pairs, we choose an anchor modality $a \in \{s, t, h, p\}$ as one of the four modalities. Therefore, for a batch of $B$ samples, the contrastive loss on their learned embeddings is defined as follows:

$$\mathcal{L}_{\text{vol}}^{\rightarrow} = -\frac{1}{B} \sum_{i=1}^{B} \log \frac{\exp(-V(a_i, f_i^t, f_i^h, f_i^p)/\tau)}{\sum_{j=1}^{B'} \exp(-V(a_j, f_i^t, f_i^h, f_i^p)/\tau)}, \tag{2}$$

where, for example, the first modality $f_i^s$ is chosen as the anchor $a_i$, negative pairs are constructed by permuting the anchor, and $\tau$ is the temperature parameter. We also add the reverse loss (w.r.t. negative pairs construction) to ensure symmetric alignment: $\mathcal{L}_{\text{vol}}^{\leftarrow} = -\frac{1}{B} \sum_{i=1}^{B} \log \frac{\exp(-V(a_i, f_i^t, f_i^h, f_i^p)/\tau)}{\sum_{j=1}^{B'} \exp(-V(a_i, f_j^t, f_j^h, f_j^p)/\tau)}$. The combined volume-based loss is

$$\mathcal{L}_{\text{vol}} = \frac{1}{2}(\mathcal{L}_{\text{vol}}^{\rightarrow} + \mathcal{L}_{\text{vol}}^{\leftarrow}) \tag{3}$$

## 3.2 GRADIENT-INFORMED ADAPTIVE MODALITY SELECTION

While volume-based contrastive loss treats all modalities equally, different modalities may vary in quality and relevance, with contributions that change during training. Static fusion strategies risk either underutilizing weaker modalities or overfitting to dominant ones. We propose a gradient-informed modality dropout mechanism that dynamically adapts modality usage based on their instantaneous contribution to the loss function.

**Gradient Contribution Analysis**   Assume $\mathcal{L}_{\tilde{t}}$ denotes mini-batch loss at training step $\tilde{t}$. We measure the importance of modality $m \in \{s, t, h, p\}$ by the magnitude of the gradient with respect to its embedding:

$$g_{\tilde{t}}^m = \left\| \frac{\partial \mathcal{L}_{\tilde{t}}}{\partial f_{\tilde{t}}^m} \right\|_2 \tag{4}$$

where $f_{\tilde{t}}^m \in \mathbb{R}^d$ is the learned embedding of modality $m$ at gradient step $\tilde{t}$. To avoid noisy decisions, we track the history of gradient contributions over the past $K$ steps: $\bar{g}_{\tilde{t}}^m = \frac{\sum_{k=0}^{K-1} \alpha^k g_{\tilde{t}-k}^m}{\sum_{k=0}^{K-1} \alpha^k}$, where $\alpha \in (0, 1)$ is an exponential decay factor which yields a smooth, temporally discounted importance score for each modality.

**Adaptive Modality Dropping Strategy** We employ a principled adaptive strategy that considers both the magnitude and variance of gradient contributions. Let $\mu_{\tilde{t}} = \frac{1}{4}\sum_m \bar{g}_{\tilde{t}}^m$ and $\sigma_{\tilde{t}} = \sqrt{\frac{1}{4}\sum_m (\bar{g}_{\tilde{t}}^m - \mu_{\tilde{t}})^2}$ denote the mean and standard deviation of weighted gradients across modalities at the current gradient step $\tilde{t}$. We will drop a modality from the volume based contrastive loss calculation with a probability of $p_{\text{drop}}$, which is a hyperparameter. The criteria to drop a modality is defined as follows:

$$
m_{\text{drop}}^{(\tilde{t})} = \begin{cases} \arg\max_m \bar{g}_{\tilde{t}}^m & \text{if dominance detected, e.g., } \bar{g}_{\tilde{t}}^m > \mu_{\tilde{t}} + \lambda_\sigma \sigma_{\tilde{t}}, \\ \arg\min_m \bar{g}_{\tilde{t}}^m & \text{otherwise,} \\ \text{none} & \text{with probability } (1 - p_{\text{drop}}). \end{cases} \tag{5}
$$

where $\lambda_\sigma = 1.5$ is the threshold multiplier. This means that we adaptively drop modalities based on two criteria: 1) *Dominance prevention*: if a modality's contribution is much larger than others, we drop it to avoid overfitting; 2) *Low-contribution pruning*: Otherwise, we drop the modality with the smallest gradient contribution to encourage use of more informative signals. This dynamic selection balances stability and diversity, ensuring all modalities remain engaged throughout training.

## 3.3 WEAK SUPERVISION THROUGH IC50 ACTIVITY MEASURE

As the IC50 values for wide range of protein-small molecule pairs are available on public data sources such as BindingDB (Gilson et al., 2016), we introduce an additional classification task as an auxiliary objective during pre-training. However, IC50 labels are not available for all possible protein-small molecule pairs, this task provides only weak supervisory signal during pre-training when IC50 information is available. We train a classifier $F_\phi^{IC50}$ to predict the IC50 class from the learned embeddings of all four modalities: $f^{\text{fused}} = [f^s; f^t; f^h; f^p] \in \mathbb{R}^{4d}$. Note that IC50 values are continuous, but given the inherent challenges of IC50 regression, including heterogeneous value distributions, wide dynamic ranges spanning several orders of magnitude, and noisy measurements (Qureshi et al., 2015; Bavi et al., 2016; Ashraf et al., 2023), we formulate the problem as a three-class classification task by employing discretizations on IC50 values (see Appendix A).

However, this discretization comes with class-imbalance described in Appendix A. To address this issue, we employ a weighted cross-entropy loss:

$$
\mathcal{L}_{\text{IC50}} = -\frac{1}{|\mathcal{S}|}\sum_{i \in \mathcal{S}} w_{y_i} \log p(y_i | f_i^{\text{fused}}), \tag{6}
$$

where $\mathcal{S}$ denotes the set of samples with valid IC50 annotations, and class weights are computed as: $w_c = \frac{N_{\text{total}}}{C \cdot N_c}$, where $N_{\text{total}}$ being the total number of samples, $C$ the number of classes, and $N_c$ the number of samples in class $c$.

**Auxiliary Bimodal Contrastive Loss** As the downstream task involves protein and molecule embeddings only, to emphasize alignment between these two, we also explicitly incorporate traditional pairwise contrastive losses between SMILES and protein modalities: $\mathcal{L}_{\text{bi}} = \frac{1}{2}(\mathcal{L}_{s \to p} + \mathcal{L}_{p \to s})$ where $\mathcal{L}_{s \to p}$ and $\mathcal{L}_{p \to s}$ follow the standard CLIP-style contrastive formulation (Radford et al., 2021).

## 3.4 UNIFIED TRAINING OBJECTIVE

The complete training objective integrates all components with appropriate weighting:

$$
\mathcal{L}_{\text{total}} = \lambda_1 \mathcal{L}_{\text{vol}} + \lambda_2 \mathcal{L}_{\text{bi}} + \lambda_3 \mathcal{L}_{\text{IC50}} \tag{7}
$$

where $\lambda_1, \lambda_2, \lambda_3$ are hyperparameters. Note that $\mathcal{L}_{\text{vol}}$ and $\mathcal{L}_{\text{bi}}$ are applied on all the training instances while $\mathcal{L}_{\text{IC50}}$ are only applied for pairs of protein and molecule with valid IC50 annotations. For gradient-based dropping of a modality in volume contrastive loss, we use $\mathcal{L} = \lambda_2 \mathcal{L}_{\text{bi}} + \lambda_3 \mathcal{L}_{\text{IC50}}$. See Appendix C for details on model architecture and parameters.

## 4 EXPERIMENTS

### 4.1 DATASET

For pre-training, we employ the multimodal molecular dataset from TRIDENT (Jiang et al., 2025), consisting of 47,269 triplets of SMILES, text descriptions, and HTA annotations. We extend this dataset by integrating protein binding information from BindingDB (Gilson et al., 2016), creating quadruplets of ⟨SMILES, Text, HTA, Protein⟩ with IC50 measurements when available. To prevent data leakage, we removed overlapping (SMILES, protein) pairs from our downstream evaluation datasets. The final pretraining dataset contains 6,545 unique molecules and 4,418 proteins, forming 50,968 quadruplets, of which 16,035 include quantitative IC50 measurements for auxiliary supervision. See Appendix B for detailed dataset construction and statistics. Ideally, we would remove any drug or target that appears in the downstream tasks from the pretraining corpus. However, given the number of downstream tasks we evaluate, this would leave too little data for effective pretraining. Consequently, we only exclude overlaps at the (SMILES, protein) pair level. To verify that our method does not memorize entity-specific patterns, we perform an overlap analysis on the Activation dataset; results are provided in appendix E.3.

We evaluated our approach on four benchmark datasets from the DTIAM framework (Lu et al., 2025). These datasets cover two types of prediction tasks: drug-target interaction (DTI) prediction using the Yamanishi_08 and Hetionet datasets, and mechanism of action (MoA) prediction using the Activation and Inhibition datasets. **1) Activation dataset** obtained from the Therapeutic Target Database (TTD) (Zhou et al., 2022), containing 1,426 drugs, 281 targets, and 1,913 known activation interactions. **2) Yamanishi_08** originally introduced by (Yamanishi et al., 2008) consists of four sub-datasets: G-Protein Coupled Receptors, Ion Channels, Nuclear Receptors, and Enzymes. We use the combined dataset constructed by (Ye et al., 2021), containing 791 drugs, 989 targets, and 5,127 known DTIs. **3) Hetionet dataset** constructed by (Himmelstein et al., 2017), which integrated biomedical data from 29 public resources, comprising 1,384 drugs, 5,763 targets, and 49,942 DTIs. **4) Inhibition dataset** derived from TTD (Zhou et al., 2022), containing 14,049 drugs, 1,088 targets, and 21,055 known inhibition interactions. For detailed dataset statistics, see Appendix Table 3.

**Pre-training** Our four-modal contrastive learning framework employs a two-stage training pipeline designed for computational efficiency and scalability. In the first stage, we extract embeddings using domain-specific pre-trained encoders: MoLFormer-XL (Ross et al., 2022) for SMILES sequences, MolT5 (Edwards et al., 2022) for textual descriptions and HTA annotations, and ESM2 (Lin et al., 2023) for protein sequences. In the second stage, we train lightweight projection networks that map these modality-specific embeddings into a unified representation space, where volume-based contrastive alignment is performed using distributed training across multiple GPUs. The complete training procedure, including our novel gradient-informed adaptive modality dropout strategy, is detailed in Algorithms 1 and 2 in the Appendix.

Notably, we deliberately exclude $\mathcal{L}_{\text{vol}}$ from the gradient computation for modality dropping. Instead, we use $\mathcal{L}_{\text{bi}}$ and $\mathcal{L}_{\text{IC50}}$ to assess modality importance for two key reasons. First, the bimodal contrastive loss and IC50 loss provide stable, interpretable signals about each modality's contribution without creating computational circularity. Second, IC50 values, though sparsely available, offer biologically meaningful supervision that directly reflects protein-molecule interaction strength, making the gradients from $\mathcal{L}_{\text{IC50}}$ particularly valuable for identifying which modalities are most important for drug-target activity prediction. Comprehensive training configuration details are provided in Appendix C. In Table 4, we present the network architecture along with the hyperparameter values used in our experiments. In Tables 6, 7, 8, and in Figure 4, we also provide the sensitivity analysis with respect to the hyperparameters.

We construct negative samples using an anchor point (Eq. 2); in each negative sample only a single modality is altered while the remaining modalities remain aligned. We hypothesize this is the most challenging negative sample scenario, since the model must distinguish the positive case, where all modalities are aligned, from a negative case in which all but one modality are aligned. To assess alternative strategies, we also evaluate an aggressive multi-domain negative sampling scheme in which negative samples are formed by varying multiple modalities simultaneously. The results are presented in Appendix section C.6.

Table 1: Mean performance comparison between GRAM-DTI and state-of-the-art baselines on DTI and MoA prediction tasks across multiple datasets and data splitting scenarios. GRAM-DTI demonstrates superior performance in most evaluation settings. † indicates reproduced results; other results are from baseline papers. **Bold** denotes best performance.

| Data | Metric | Scenario | CPL-GNN | MPNN-CNN | TransformerCPI | KGE-NFM | DTIAM † | GRAM-DTI | Data | AI-DTI | DTIAM † | GRAM-DTI |
|---|---|---|---|---|---|---|---|---|---|---|---|---|
| Yamanishi_08 | AUPR | Warm start | 0.431 | 0.816 | 0.802 | 0.817 | 0.901±0.0085 | **0.904±0.0079** | Activation | 0.583 | 0.623±0.0245 | **0.642±0.0221** |
| | | Drug cold start | 0.167 | 0.408 | 0.410 | 0.341 | 0.439±0.0580 | **0.440±0.0662** | | 0.550 | 0.611±0.0252 | **0.628±0.0222** |
| | | Target cold start | 0.380 | 0.602 | 0.646 | 0.761 | 0.844±0.0350 | **0.849±0.0312** | | 0.219 | 0.391±0.0320 | **0.450±0.0374** |
| | AUROC | Warm start | 0.821 | 0.952 | 0.953 | 0.948 | 0.967±0.0050 | **0.977±0.0042** | | 0.888 | 0.903±0.0088 | **0.914±0.0078** |
| | | Drug cold start | 0.629 | 0.797 | 0.767 | 0.779 | 0.818±0.0255 | **0.828±0.0285** | | 0.879 | 0.907±0.0076 | **0.913±0.0068** |
| | | Target cold start | 0.800 | 0.856 | 0.870 | 0.923 | 0.941±0.0180 | **0.955±0.0155** | | 0.652 | 0.792±0.0240 | **0.834±0.0258** |
| Hetionet | AUPR | Warm start | 0.441 | 0.734 | - | 0.789 | **0.879±0.0095** | 0.859±0.0082 | Inhibition | 0.840 | **0.845±0.0070** | 0.785±0.0061 |
| | | Drug cold start | 0.219 | 0.453 | - | 0.391 | 0.514±0.0680 | **0.529±0.0626** | | 0.830 | 0.731±0.0045 | 0.756±0.0034 |
| | | Target cold start | 0.433 | 0.470 | - | **0.651** | 0.625±0.0210 | 0.626±0.0239 | | 0.215 | 0.445±0.0620 | **0.464±0.0559** |
| | AUROC | Warm start | 0.810 | 0.956 | - | 0.968 | 0.957±0.0015 | **0.981±0.0011** | | 0.952 | **0.954±0.0025** | 0.949±0.0018 |
| | | Drug cold start | 0.685 | 0.831 | - | 0.803 | 0.752±0.0355 | **0.855±0.0385** | | **0.948** | 0.921±0.0028 | 0.940±0.0018 |
| | | Target cold start | 0.810 | 0.858 | - | 0.915 | 0.917±0.0090 | **0.921±0.0079** | | 0.605 | 0.819±0.0205 | **0.823±0.0028** |

**Downstream task**  In the DTI and MoA prediction task, the objective is to determine whether a given drug-target pair interacts, which constitutes a binary classification problem. Note that existing datasets only include those pairs that interacts (positive class). Following standard practice (Lu et al., 2025), we generated negative samples using a 1:10 ratio with positive samples for all datasets. To evaluate the model's generalization performance, we employed three different data splitting strategies for train-test division: 1) *warm start*: The data is split based on protein-molecule pairs, ensuring that no common pairs appear in both the training and test sets. 2) *drug cold start*: This split is performed at the molecule level, guaranteeing that no drug in the test set is present in the training set. 3) *target cold start*: Similar to the above, but split at the protein level, meaning no protein in the test set is seen during training. These three settings allow us to assess how well the model performs when faced with unseen molecule-protein pairs, unseen molecules, or unseen proteins, respectively. For evaluation, we followed the cross-validation protocols established in the original DTIAM framework (Lu et al., 2025): 10-fold cross-validation for DTI prediction tasks (Yamanishi_08 and Hetionet datasets) and 5-fold cross-validation for MoA prediction tasks (Activation and Inhibition datasets). Note that we generated negative samples at a 1:10 ratio relative to positive samples across all datasets, to ensure consistency with baseline methods and a fair comparison. Additional results with varying negative-sample ratios are provided in Appendix E.4, illustrating how our model performance changes as the ratio is adjusted.

## 4.2 EXPERIMENTAL RESULTS

We evaluated GRAM-DTI against state-of-the-art models across multiple benchmark datasets to demonstrate its effectiveness. For DTI prediction tasks, Table 1 presents a comparison with five baselines: CPL-GNN (Tsubaki et al., 2019), MPNN-CNN (Gilmer et al., 2017), TransformerCPI (Chen et al., 2020), and KGE-NFM (Ye et al., 2021) and DTIAM (Lu et al., 2025), on the Yamanishi_08 and Hetionet datasets. For MoA prediction tasks, we compared GRAM-DTI against two baselines: AI-DTI (Lee et al., 2023) and DTIAM (Lu et al., 2025) on the Activation and Inhibition datasets. The different baseline sets reflect the distinct methodological approaches and evaluation standards established for DTI and MoA prediction in the computational drug discovery community and follows prior works (Lu et al., 2025; Panahandeh & Mansouri, 2025).

GRAM-DTI demonstrates strong performance across benchmark datasets, with particularly notable gains in target cold start scenarios. For DTI tasks, our method achieves substantial improvements on Yamanishi_08 in both warm start and target/drug cold start settings. On the larger Hetionet dataset, GRAM-DTI outperforms most baselines across multiple evaluation scenarios. For MoA prediction, GRAM-DTI consistently surpasses baselines on the Activation dataset, especially under target cold start conditions. On the Inhibition dataset, while GRAM-DTI does not outperform existing baselines in warm start and drug cold start settings, it exhibits excellent performance in target cold start.

Overall, GRAM-DTI outperforms state-of-the-art baselines in nearly all evaluation settings—10 out of 12 for DTI and 8 out of 12 for MoA tasks. Its strongest gains emerge on smaller datasets (Yamanishi_08 and Activation), where pre-training provides the greatest benefit under limited supervision, thus validating its potential for real-world drug discovery applications with limited available labeled data. On larger datasets (Hetionet and Inhibition), GRAM-DTI remains on par with or outperforms strong baselines, particularly in cold start conditions. These results highlight the robustness and generalizability of our multimodal alignment framework, especially when extending to novel proteins.

Table 2: Zero-shot retrieval performance comparison between GRAM-DTI and DTIAM baseline across four datasets. Results show Recall@K metrics for bidirectional retrieval tasks: S→P and P→S. GRAM-DTI demonstrates superior retrieval capability across most scenarios and datasets using only pretrained representations. **Bold** denotes best performance.

| Direction | Metric | Yamanishi_08 | | Hetionet | | Activation | | Inhibition | |
|---|---|---|---|---|---|---|---|---|---|
| | | DTIAM | GRAM-DTI | DTIAM | GRAM-DTI | DTIAM | GRAM-DTI | DTIAM | GRAM-DTI |
| S→P | R@1 | 0.0038±0.0004 | **0.0465±0.0027** | 0.0043±0.0002 | **0.0331±0.0038** | 0.0028±0.0002 | **0.0136±0.0011** | 0.0004±0.0000 | **0.0055±0.0003** |
| | R@10 | 0.0341±0.0042 | **0.1691±0.0084** | 0.0434±0.0051 | **0.1340±0.0025** | 0.0266±0.0037 | **0.1020±0.0067** | 0.0097±0.0006 | **0.0337±0.0011** |
| | R@100 | 0.1960±0.0181 | **0.4449±0.0075** | 0.2066±0.0109 | **0.3616±0.0063** | 0.3184±0.0229 | **0.5688±0.0172** | 0.1036±0.0104 | **0.1994±0.0018** |
| P→S | R@1 | 0.0040±0.0002 | **0.0742±0.0120** | **0.0404±0.0028** | 0.0236±0.0010 | 0.0071±0.0008 | **0.0370±0.0069** | 0.0000±0.0000 | **0.0221±0.0061** |
| | R@10 | 0.0849±0.0089 | **0.2465±0.0256** | **0.1319±0.0095** | 0.1049±0.0055 | 0.0463±0.0050 | **0.2454±0.0142** | 0.0028±0.0004 | **0.0819±0.0065** |
| | R@100 | 0.3670±0.0186 | **0.5540±0.0148** | 0.3632±0.0474 | **0.3841±0.0082** | 0.2206±0.0264 | **0.6029±0.0231** | 0.0588±0.0049 | **0.2325±0.0094** |

## 4.3 ZERO-SHOT RETRIEVAL TASK

In addition to predicting drug–target interactions, an important aspect of evaluating our model's effectiveness is its ability to accurately retrieve relevant molecules or proteins based on a given query. The retrieval task assesses the model's capacity to learn meaningful, high-quality representations that preserve semantic relationships across different modalities. This task is particularly relevant for applications such as drug repurposing and target identification (Luo et al., 2016; Pushpakom et al., 2019), where retrieving similar compounds or proteins can guide experimental validation and discovery.

To evaluate the retrieval capability of GRAM-DTI, we conduct a series of experiments across the same four datasets. For each dataset, we formulate two retrieval scenarios: (i) retrieving proteins given a drug query (S→P), and (ii) retrieving drugs given a protein query (P→S). Using the learned representations directly from our pre-training framework without any additional training, we compute similarity scores between query and candidate items. The performance is measured using standard metrics, including Recall@K (R@1, R@10, R@100), which indicate the proportion of relevant items retrieved within the top-K results.

The results, summarized in Table 2, demonstrate that our method outperforms DTIAM (the best baseline from DTI and MoA experiments in Table 1) across nearly all datasets and metrics. Notably, the superior performance in R@1 and R@10 indicates that our model effectively captures the semantic relationships necessary for accurate retrieval, highlighting the quality of the learned multimodal representations. These strong zero-shot retrieval results provide compelling evidence that our multimodal pretraining framework successfully learns meaningful drug-target representations that generalize well beyond the specific downstream prediction tasks. More detailed experimental results can be found in Appendix E.

## 4.4 ABLATION STUDY

Note that our main pre-training objective consists of three components (see Eq.7). To evaluate the contribution of each component, we conducted a comprehensive ablation study, comparing the performance of our model with each component systematically removed.

We conduct five ablation experiments to evaluate the contribution of each component. **Exp 1** uses the full objective with modality dropout applied on volume loss calculation, i.e., $\mathcal{L} = \mathcal{L}_{\text{total}}$, which is the same as our GRAM-DTI setup. **Exp 2** pre-trains without volume loss, using $\mathcal{L} = \lambda_2\mathcal{L}_{\text{bi}} + \lambda_3\mathcal{L}_{\text{IC50}}$. **Exp 3** pre-trains without traditional pairwise contrastive loss, employing $\mathcal{L} = \lambda_1\mathcal{L}_{\text{vol}} + \lambda_3\mathcal{L}_{\text{IC50}}$. **Exp 4** pre-trains without IC50 supervision, using $\mathcal{L} = \lambda_1\mathcal{L}_{\text{vol}} + \lambda_2\mathcal{L}_{\text{bi}}$. Finally, **Exp 5** uses the full objective but without modality dropout. The ablation study results on Activation dataset is presented in Figure 2 while the same for Yamanishi_08 dataset is reported in Appendix Figure 5. Across all setups, the full GRAM-DTI model (Exp 1) with all components enabled generally outperforms other variants where one component is removed.

**Impact of Gramian Volume-Based Alignment.** Gramian volume-based alignment provides substantial benefits across most evaluation scenarios. Comparing it (Exp 1) with the variant excluding volume loss (Exp 2) reveals consistent improvements across the majority of metrics, particularly in challenging scenarios like target cold start where models must generalize to previously unseen proteins. The volume-based approach effectively captures higher-order relationships among the four modalities that cannot be achieved through pairwise alignments alone, leading to more robust multimodal representations.

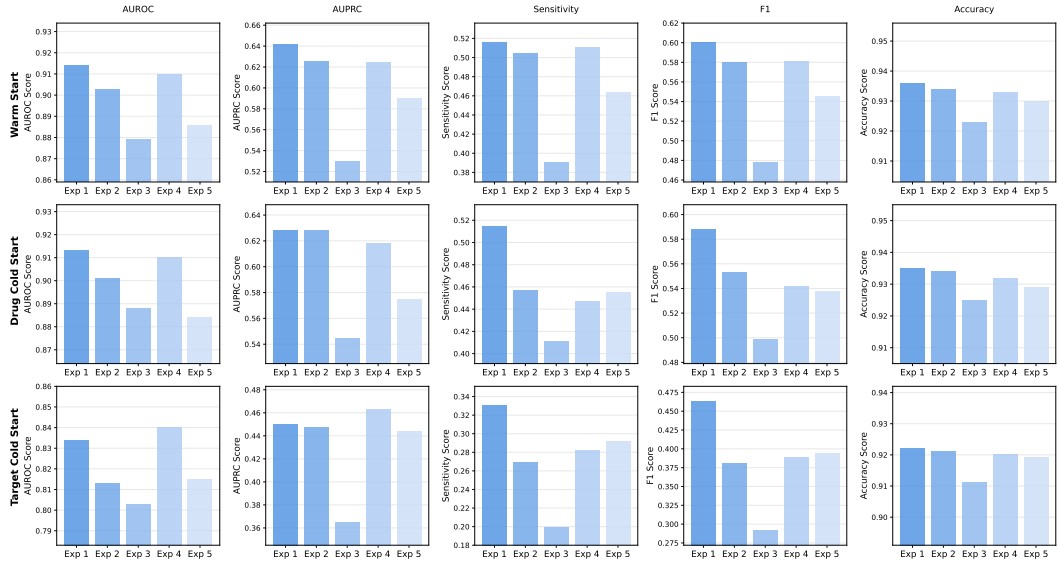

Figure 2: Ablation study results on the Activation dataset across five experimental configurations and three data splitting scenarios. The full GRAM-DTI model (Exp 1) outperforms variants with removed components in most cases, demonstrating the synergistic contribution of each training objective component.

**Impact of IC50 Auxiliary Supervision and Contrastive Loss.** Incorporating IC50 auxiliary supervision consistently improves performance across most evaluation scenarios (with the exception of Activation target cold start) as seen by comparing Exp 1 with Exp 4 (without IC50 supervision). Same conclusion holds when comparing Exp 1 with Exp 3, which suggests that the bimodal contrastive loss also ensures robust drug-protein alignment and complements volume-based alignment. Together, these components capture both molecular activity principles and critical drug-protein relationships for effective prediction.

**Impact of Adaptive Modality Dropout.** Removing adaptive modality dropout (Exp 5), we see in figure 2, the performance consistently deteriorates, often by a large margin, compared to the no-dropout setting. By dynamically regulating modality contributions during training, the adaptive dropout prevents dominant modalities from overwhelming complementary signals while ensuring all modalities remain engaged. This prevents overfitting to specific modality combinations, ultimately leading to more generalizable representations. To further validate this design choice, we compared our probabilistic dropout strategy against "soft" weighting alternatives (e.g., weighted-modality gradients). Results (see Appendix E.6) demonstrate that our "hard" dropout strategy provides a stronger regularization effect and superior downstream performance.

**Impact of Molecular Encoder Strength.** To assess the modularity of our framework, we evaluated GRAM-DTI using more advanced molecular encoders, specifically Uni-Mol2 (Ji et al., 2024) and BioT5+ (Pei et al., 2024), in place of MolFormer. As detailed in Appendix E.7, we observe that stronger encoders yield further performance gains, confirming that GRAM-DTI effectively leverages improvements in upstream foundation models.

**Multimodal Embedding Evolution** To visualize how GRAM-DTI learns unified representations, we examine embedding evolution across training epochs using t-SNE on 3,000 randomly sampled quadruplets (Figure 3). Initially, the four modalities form distinct, separate clusters. As training progresses, volume-based alignment gradually transforms rigid modality boundaries into semantically integrated representations while preserving modality-specific structures. By epoch 40, embeddings show substantial cross-modal integration where instances cluster by semantic relationships rather than purely by modality type. This evolution pattern provides visual evidence that our approach successfully balances cross-modal alignment with modality-specific information retention, supporting the quantitative improvements observed in downstream tasks. Additional analyses with varying sample sizes are provided in Appendix F.1.

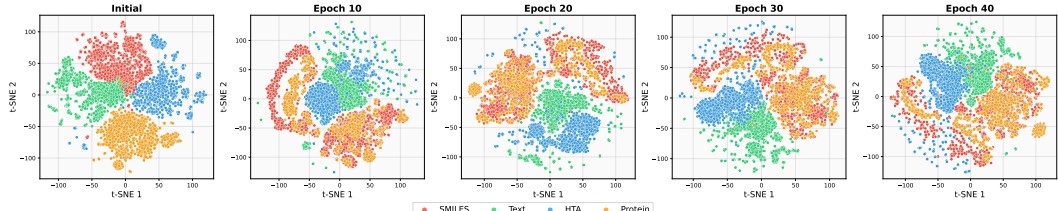

Figure 3: Evolution of multimodal embeddings during GRAM-DTI pre-training visualized using t-SNE on 3,000 samples. Four modalities (SMILES, Text, HTA, Protein) progressively align from separate clusters to semantically integrated representations, demonstrating effective volume-based multimodal alignment.

**Impact of missing modalities during pre-training**    Bringing in as many relevant modalities in the pretraining would help learn better representation for the corresponding downstream task. However, what if during training, not all the modalities are available? We investigated this question by considering scenario where certain modality is not available during pretraining. The results are presented in Appendix section E.2. Moreover, we can extend pretraining to include samples with missing modalities, which would substantially increase the size of our training set. To assess whether all modalities are beneficial, our current pretraining phase includes only samples in which all four modalities are present, a choice that significantly limits the dataset. As a proof of concept, we evaluated whether including samples with only a subset of modalities improves downstream performance. The results, presented in Appendix E.5, indicate a promising direction: incorporating partial-modality samples can expand the pretraining corpus and may enhance model performance.

### 4.5    FALSE NEGATIVE CASE ANALYSIS

To understand better when the model fails to predict drug target activity, we first systematically identified the top 10 "hardest" false negatives in the Activation dataset—pairs where the model predicted a strong negative signal despite a positive ground truth label. These are listed in table 22 in the appendix section E.8. From this list, we performed a detailed case study on Rank: Drug D03XIS (R-568) targeting T92076 (CASR). Our analysis suggests this prediction difficulty likely stems from the unique and complex biology of this pair, which is statistically rare in typical drug-target datasets: This case may be challenging because CASR is a Class C GPCR, fundamentally different from the Class A GPCRs that dominate drug databases. Three key factors may contribute to the prediction difficulty: (1) CASR has a large extracellular Venus flytrap domain, contrasting with the compact transmembrane binding pockets typical of Class A GPCRs; (2) it functions as an obligate homodimer with complex inter-protomer allosteric signaling; (3) R-568 acts as a positive allosteric modulator rather than a traditional orthosteric agonist. The prediction difficulty may reflect the biological rarity of Class C GPCR allosteric modulators in drug discovery.

## 5    CONCLUSION

We presented GRAM-DTI, a multimodal pretraining framework that extends volume-based contrastive learning to four modalities with gradient-informed adaptive modality dropout and IC50 auxiliary supervision. Evaluation across four benchmark datasets shows GRAM-DTI consistently outperforms baselines, particularly in cold start scenarios. Ablation studies (Appendix section 4.4) confirm synergistic contributions of each component. These results highlight the potential of multimodal pretraining for drug discovery, where integrating diverse data sources leads to more robust prediction models. Currently, the need to construct complete quadruplets ⟨SMILES, Text, HTA, Protein⟩ and remove overlapping (protein, SMILES) pairs with the downstream task has limited the scale of our pre-training dataset, restricting the diversity of molecules and proteins. To fully unlock the potential of GRAM-DTI and improve generalization to unseen molecular and protein targets, expanding the pre-training corpus will be crucial. In addition, incorporating protein-related modalities beyond sequence information could further enhance performance.

### ACKNOWLEDGEMENTS

This work was partially supported by a Johnson & Johnson grant for LLM-based toxicity prediction, US National Science Foundation IIS-2412195, CCF-2400785, the Cancer Prevention and Research Institute of Texas (CPRIT) award (RP230363) and the National Institutes of Health (NIH) R01 award (1R01AI190103-01).

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

## A    IC50 VALUES DISCRETIZATIONS

Given the inherent challenges of IC50 regression—including heterogeneous value distributions, wide dynamic ranges spanning several orders of magnitude, and noisy measurements—we formulate the problem as a three-class classification task. The IC50 values are discretized based on pharmaceutical relevance thresholds:

$$\text{IC50 class} = \begin{cases} 0 & \text{if IC50} < 10\mu\text{M (effective)} \\ 1 & \text{if } 10\mu\text{M} \leq \text{IC50} \leq 1000\mu\text{M (moderate)} \\ 2 & \text{if IC50} > 1000\mu\text{M (ineffective)} \end{cases} \tag{8}$$

This discretization strategy aligns with established drug discovery practices (Qureshi et al., 2015; Bavi et al., 2016; Ashraf et al., 2023) where compounds with IC50 $< 10\mu$M are considered highly active, those between $10 - 1000\mu$M show moderate activity, and those $> 1000\mu$M are typically considered inactive.

## B    DATASET

**Pretraining Data**    Our pretraining dataset builds upon the high-quality multimodal molecular dataset from TRIDENT (Jiang et al., 2025), which provides comprehensive molecular representations through the integration of SMILES strings, natural language descriptions, and Hierarchical Taxonomic Annotations (HTA). The original TRIDENT dataset contains 47,269 carefully curated ⟨SMILES, Text, HTA⟩ triplets sourced from PubChem, where each molecule is annotated across 32 diverse taxonomic classification systems.

To enable protein-molecule interaction modeling, we extended this dataset by incorporating binding affinity information from BindingDB, a comprehensive database of measured binding affinities for protein-molecule interactions. We mapped molecules from the TRIDENT dataset to BindingDB entries using molecular identifiers, creating 5-tuples of the form ⟨SMILES, Text, HTA, Protein, IC50⟩. This integration combines the rich semantic and structural information from TRIDENT with quantitative binding affinity measurements, providing a unified multimodal representation that captures both molecular properties and protein-molecule interactions. Following standard practices in molecular property prediction, we implemented careful data filtering to prevent information leakage between pretraining and downstream evaluation. Specifically, we removed all SMILES-protein binding pairs that appear in our downstream task datasets to ensure fair evaluation and prevent overfitting to specific molecular-protein combinations seen during pretraining.

After filtering, 6,545 unique molecules have associated protein binding information. Considering that each molecule can interact with multiple proteins, this results in a total of 50,968 quadruplets ⟨Protein, SMILES, Text, HTA⟩, covering 4,418 unique proteins. Among these quadruplets, 16,035 entries include quantitative IC50 measurements, providing high-quality binding affinity annotations for modeling.

**Downstream Task Datasets**    We evaluated our approach on four benchmark datasets (see Table 3) from the DTIAM framework (Lu et al., 2025), covering drug-target interaction (DTI) prediction and mechanism of action (MoA) prediction tasks. 1) **Activation dataset** obtained from the Therapeutic Target Database (TTD) (Zhou et al., 2022), containing 1,426 drugs, 281 targets, and 1,913 known activation interactions. 2) **Yamanishi_08** originally introduced by (Yamanishi et al., 2008) and consists of four sub-datasets: G-Protein Coupled Receptors (GPCR), Ion Channels (IC), Nuclear Receptors (NR), and Enzymes (E). We use the combined dataset constructed by (Ye et al., 2021), containing 791 drugs, 989 targets, and 5,127 known DTIs. 3) **Hetionet dataset** constructed by (Himmelstein et al., 2017), which integrated biomedical data from 29 public resources, comprising 1,384 drugs, 5,763 targets, and 49,942 DTIs. 4) **Inhibition dataset** also derived from TTD (Zhou et al., 2022), containing 14,049 drugs, 1,088 targets, and 21,055 known inhibition interactions.

The MoA refers to how a drug works on its target to produce the desired effects, which involve two major roles: activation and inhibition mechanisms. Distinguishing the activation and inhibition MoA between drugs and targets is critical and challenging in the drug discovery and development process, as well as their clinical applications Zhang et al. (2023).

Table 3: Statistics of downstream task datasets for binary classification. Known Interactions represents the number of positive drug-target binding pairs, while Total Samples includes both positive samples and 10 times negative samples generated following standard practice.

| Dataset | Task Type | Drugs | Targets | Known Interactions | Total Samples |
|---|---|---|---|---|---|
| Yamanishi_08 | DTI | 791 | 989 | 5,127 | 56,397 |
| Hetionet | DTI | 1,384 | 5,763 | 49,942 | 549,362 |
| Activation | MoA | 1,426 | 281 | 1,913 | 21,043 |
| Inhibition | MoA | 14,049 | 1,088 | 21,055 | 231,605 |

## C  PRE-TRAINING SETUP AND ARCHITECTURAL DETAILS

### C.1  PRE-TRAINING INFRASTRUCTURE

Our four-modal contrastive learning framework employs a two-stage training pipeline. First, we extract embeddings from domain-specific pre-trained models (MoLFormer-XL (Ross et al., 2022) for SMILES, MolT5(Edwards et al., 2022) for text/HTA, ESM2 (Lin et al., 2023) for proteins). Second, we train projection networks and the GRAM4Modal loss using distributed training across multiple GPUs. The complete training procedure is detailed in Algorithm 1, which incorporates our gradient-based modality dropping strategy (Algorithm 2).

Notably, we deliberately exclude $\mathcal{L}_{\text{vol}}$ from the gradient computation for modality dropping to avoid circular dependency, where the volume loss computation would depend on gradients derived from that same computation. Instead, we use $\mathcal{L} = \lambda_2 \mathcal{L}_{\text{bi}} + \lambda_3 \mathcal{L}_{\text{IC50}}$ to assess modality importance for two key reasons: 1) *Avoiding circular dependency*: The bimodal contrastive loss and IC50 loss provide stable, interpretable signals about each modality's contribution without creating computational circularity; 2) *Leveraging weak supervision*: IC50 values, though sparsely available, offer biologically meaningful supervision that directly reflects protein-molecule interaction strength. The gradients from $\mathcal{L}_{\text{IC50}}$ thus provide valuable information about which modalities are most important for predicting drug-target activity, making them suitable signals for adaptive modality selection. Table 4 provides comprehensive training configuration details.

---

**Algorithm 1** Four-Modal Contrastive Learning with Gradient-based Modality Dropping

---

**Require:** Pre-computed embeddings $\{x_i^s, x_i^t, x_i^h, x_i^p\}$
**Require:** Drop probability $p_{\text{drop}}$, temperature $\tau$
**Ensure:** Projected features $\{f^s, f^t, f^h, f^p\}$
1: $f^m \leftarrow F_\phi^m(E_m(x^m))$ for $m \in \{s, t, h, p\}$
2: $f^m \leftarrow \|f^m\|_2 = 1$ for all modalities
3: $d \leftarrow \text{GradientBasedDrop}(\{f^m\}, \mathcal{L}, p_{\text{drop}})$
4: **if** $d.\text{should\_drop} = \text{False}$ **then**
5:   $V_f \leftarrow \text{GRAM4Modal}(f^p, \{f_{\text{all}}^s, f_{\text{all}}^t, f_{\text{all}}^h\})$
6:   $V_r \leftarrow \text{GRAM4Modal}(f_{\text{all}}^p, \{f^s, f^t, f^h\})^T$
7: **else**
8:   $m_a \leftarrow d.\text{anchor\_modality}$
9:   $\{m_1, m_2\} \leftarrow \text{remaining\_modalities} \setminus \{m_a\}$
10:   $V_f \leftarrow \text{GRAM3Modal}(f^{m_a}, \{f_{\text{all}}^{m_1}, f_{\text{all}}^{m_2}\})$
11:   $V_r \leftarrow \text{GRAM3Modal}(f_{\text{all}}^{m_a}, \{f^{m_1}, f^{m_2}\})^T$
12: **end if**
13: $S_f \leftarrow -V_f/\tau, S_r \leftarrow -V_r/\tau$
14: $\mathcal{L}_{\text{vol}} \leftarrow \frac{1}{2}[\mathcal{L}_{\text{vol}}^{\rightarrow} + \mathcal{L}_{\text{vol}}^{\leftarrow}]$
15: **return** $\mathcal{L}_{\text{total}} = \lambda_1 \mathcal{L}_{\text{vol}} + \lambda_2 \mathcal{L}_{\text{bi}} + \lambda_3 \mathcal{L}_{\text{IC50}}$

---

## C.2 MODEL ARCHITECTURE

The projection networks $F_\phi^m$ map pre-computed embeddings to a unified 512-dimensional space. Each projection consists of three linear layers with GELU activations, layer normalization, and dropout (rate=0.1). The IC50 classification head $F_\phi^{IC50}$ concatenates all four modality features $f^{\text{fused}} = [f^s; f^t; f^h; f^p]$ and predicts binding affinity classes through a two-layer MLP with dropout (rate=0.3). The pre-trained encoder specifications are detailed in Table 5. All encoders $E_m$ are frozen during training to leverage their pre-trained representations while only fine-tuning the projection networks $F_\phi^m$ for computational efficiency.

Table 4: Training Configuration Parameters

| Parameter | Configuration |
|---|---|
| Hardware | Multi-GPU NVIDIA (CUDA) |
| Training framework | PyTorch DDP, NCCL |
| Batch size | 1280 per GPU |
| Learning rate | $1 \times 10^{-4}$ (Adam) |
| Epochs | 40 |
| Temperature $\tau$ | 0.07 |
| Drop probability $p_{\text{drop}}$ | 0.8 |
| Gradient history length $K$ | 5 |
| Decay factor $\alpha$ | 0.9 |
| Threshold multiplier $\lambda_\sigma$ | 1.5 |
| Loss weights $\lambda_1, \lambda_2, \lambda_3$ | 1.0, 1.0, 1.0 |
| Label smoothing | 0.1 |

---

**Algorithm 2** Gradient-based Adaptive Modality Dropping

**Require:** Features $\{f^m\}_{m \in \{s,t,h,p\}}$, current loss $\mathcal{L}_{\tilde{t}}$, drop probability $p_{\text{drop}}$
**Require:** Gradient history length $K$, decay factor $\alpha$, threshold $\lambda_\sigma = 1.5$
**Ensure:** Drop decision {should_drop, $m_{\text{drop}}$, anchor_modality}
1: **if** random() $> p_{\text{drop}}$ or not training **then**
2:     **return** {False, none, protein}
3: **end if**
4: **for** $m \in \{s,t,h,p\}$ **do**
5:     $g_{\tilde{t}}^m \leftarrow \left\| \frac{\partial \mathcal{L}_{\tilde{t}}}{\partial f_{\tilde{t}}^m} \right\|_2$
6:     Update gradient history for modality $m$
7: **end for**
8: **for** $m \in \{s,t,h,p\}$ **do**
9:     $\bar{g}_{\tilde{t}}^m \leftarrow \frac{\sum_{k=0}^{K-1} \alpha^k g_{\tilde{t}-k}^m}{\sum_{k=0}^{K-1} \alpha^k}$
10: **end for**
11: $\mu_{\tilde{t}} \leftarrow \frac{1}{4} \sum_m \bar{g}_{\tilde{t}}^m$, $\sigma_{\tilde{t}} \leftarrow \sqrt{\frac{1}{4} \sum_m (\bar{g}_{\tilde{t}}^m - \mu_{\tilde{t}})^2}$
12: **for** $m \in \{s,t,h,p\}$ **do**
13:     **if** $\bar{g}_{\tilde{t}}^m > \mu_{\tilde{t}} + \lambda_\sigma \sigma_{\tilde{t}}$ **then**
14:         $m_{\text{drop}}^{(\tilde{t})} \leftarrow m$; **break**
15:     **end if**
16: **end for**
17: **if** $m_{\text{drop}}^{(\tilde{t})}$ not found **then**
18:     $m_{\text{drop}}^{(\tilde{t})} \leftarrow \arg\min_m \bar{g}_{\tilde{t}}^m$
19: **end if**
20: $m_{\text{anchor}} \leftarrow \text{random\_choice}(\{s,t,h,p\} \setminus \{m_{\text{drop}}^{(\tilde{t})}\})$
21: **return** {True, $m_{\text{drop}}^{(\tilde{t})}$, $m_{\text{anchor}}$}

---

Table 5: Pre-trained Encoder Specifications

| Modality | Model $E_m$ | Output Dim |
|---|---|---|
| SMILES ($x^s$) | MoLFormer-XL-both-10pct | 768 |
| Text ($x^t$) | MolT5-base | 768 |
| HTA ($x^h$) | MolT5-base (shared) | 768 |
| Protein ($x^p$) | ESM2_t33_650M_UR50D | 1280 |

## C.3 COMPUTATIONAL EFFICIENCY

Our method is highly efficient because we freeze the large encoder backbones (ESM2, MolFormer, MolT5) and only train the lightweight projection layers. This significantly reduces computational and memory overhead.

- Hardware: All experiments were conducted on a single A6000 GPU.
- Peak Memory: The peak GPU memory usage during pretraining is only 0.12 GB.
- Batch Size: We use a large batch size of 1280.
- Pretraining Speed: Each pretraining epoch takes approximately 3 seconds to complete.

This demonstrates that our method is not memory-intensive and is computationally very efficient.

## C.4 HYPERPARAMETER TUNING AND SENSITIVITY ANALYSIS

We tuned the hyperparameters for the pretrained model and eventually set the final optimal values as: $\lambda_\sigma = 1.5, \lambda_1 = 1, \lambda_2 = 1, \lambda_3 = 1, K = 5$ as presented in table 4. To further analysis the model performance sensitivity with respect to each parameters, we investigated the impact of the each parameters value on the downstream task. In table 6, 7 and 8 we show the result on the Activation dataset when we change the parameters value from the optimal values that are used in our final model. These results demonstrate that our model's performance is stable within a reasonable range of these hyperparameters, with our chosen settings providing a robust and effective performance.

Table 6: Downstream-task performance on the Activation dataset: sensitivity to changes in hyperparameter values from the optimal setting, evaluated under the Warm-start setup. The row in bold indicates the optimal hyperparameter values used during pretraining and the corresponding downstream performance.

| $\lambda_1$ | $\lambda_2$ | $\lambda_3$ | gradient_std_multiplier $\lambda_\sigma$ | gradient_history_length $K$ | AUPRC | AUROC |
|---|---|---|---|---|---|---|
| **1** | **1** | **1** | **1.5** | **5** | **0.6424** | **0.9142** |
| 0.5 | 1 | 1 | 1.5 | 5 | 0.6449 | 0.9125 |
| 1 | 0.5 | 1 | 1.5 | 5 | 0.6237 | 0.9175 |
| 1 | 1 | 0.5 | 1.5 | 5 | 0.6326 | 0.9102 |
| 1 | 1 | 1 | 2 | 5 | 0.6340 | 0.9014 |
| 1 | 1 | 1 | 1.5 | 10 | 0.6155 | 0.9130 |

Table 7: Downstream-task performance on the Activation dataset: sensitivity to changes in hyperparameter values from the optimal setting, evaluated under the Drug cold start setup.

| $\lambda_1$ | $\lambda_2$ | $\lambda_3$ | gradient_std_multiplier | gradient_history_length | AUPRC | AUROC |
|---|---|---|---|---|---|---|
| **1** | **1** | **1** | **1.5** | **5** | **0.6278** | **0.9125** |
| 0.5 | 1 | 1 | 1.5 | 5 | 0.6404 | 0.9129 |
| 1 | 0.5 | 1 | 1.5 | 5 | 0.6299 | 0.9107 |
| 1 | 1 | 0.5 | 1.5 | 5 | 0.6152 | 0.9114 |
| 1 | 1 | 1 | 2 | 5 | 0.6183 | 0.9009 |
| 1 | 1 | 1 | 1.5 | 10 | 0.6156 | 0.9103 |

Table 8: Downstream-task performance on the Activation dataset: sensitivity to changes in hyper-parameter values from the optimal setting, evaluated under the Target cold start setup.

| $\lambda_1$ | $\lambda_2$ | $\lambda_3$ | gradient_std_multiplier | gradient_history_length | AUPRC | AUROC |
|---|---|---|---|---|---|---|
| **1** | **1** | **1** | **1.5** | **5** | **0.4497** | **0.8335** |
| 0.5 | 1 | 1 | 1.5 | 5 | 0.4394 | 0.8224 |
| 1 | 0.5 | 1 | 1.5 | 5 | 0.4274 | 0.8190 |
| 1 | 1 | 0.5 | 1.5 | 5 | 0.4571 | 0.8273 |
| 1 | 1 | 1 | 2 | 5 | 0.4286 | 0.8270 |
| 1 | 1 | 1 | 1.5 | 10 | 0.4343 | 0.8295 |

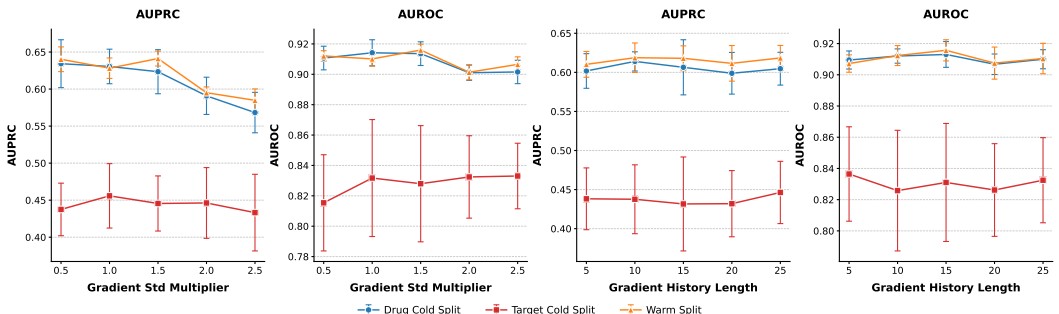

Figure 4: Effect of gradient optimization parameters on model performance. AUPRC and AUROC scores are shown for varying gradient standard deviation multiplier (left) and gradient history length (right) across three data split scenarios: drug cold split, protein (drug) cold split, and warm split. Error bars indicate standard deviation.

To further demonstrate the impact of the gradient_std_multiplier and gradient_history_length parameters on model performance, we conducted a sensitivity analysis. We fixed all other parameters and varied either gradient_std_multiplier or gradient_history_length to observe how performance changed. The sensitivity graphs are shown in Figure 4.

For the "Activation" dataset:

- gradient_std_multiplier: Increasing this parameter until a certain range had a stable effect on performance. Beyond this point, AUPRC for warm and drug cold starts began to drop, while protein cold start AUPRC remained stable. Simultaneously, AUROC for drug and warm starts decreased, while protein cold start AUROC increased.

- gradient_history_length: Model performance was relatively stable with respect to its value increase across all evaluation setups.

### C.5    VOLUME COMPUTATION DETAILS

The GRAM4Modal and GRAM3Modal functions compute volumes using Gram matrix determinants. For anchor features $f^a$ and target features $\{f^{t_1}, f^{t_2}, f^{t_3}\}$, the 4×4 Gram matrix $G$ has entries $G_{kj} = \langle f^k, f^j \rangle$. The volume is computed as $V = \sqrt{|\det(G)|}$, then converted to similarity via negative volume scaling: $S = -V/\tau$.

Algorithm 2 implements our gradient-informed adaptive modality selection strategy, which maintains consistency between forward $\mathcal{L}_{\text{vol}}^{\rightarrow}$ and reverse $\mathcal{L}_{\text{vol}}^{\leftarrow}$ contrastive computations by using a single drop decision per forward pass.

### C.6    NEGATIVE SAMPLING STRATEGIES

We construct negative samples by fixing all but one modality, producing hard negatives in which only a single modality is mismatched while the remaining modalities are aligned. This single-

modality perturbation yields a more challenging learning signal, as the model must distinguish the fully aligned positive case from near-aligned negatives. To assess alternative strategies, we also evaluate an aggressive multi-domain negative-sampling scheme in which multiple modalities are perturbed simultaneously. Specifically, for each sample $i$ in batch $B$, we generate negative samples by permuting all modalities. Results on the Activation dataset comparing cross-negative sampling to the current strategy under the volume-loss pretraining setting are reported in Table 9 and Table 10.

Table 9: Model performance comparison on activation dataset in terms of AUROC when using different negative sample strategies: Cross-Negative vs Current-Negative (Mean±Std).

| Split Type | Cross-Negative | Current-Negative |
|---|---|---|
| Warm start | 0.9142±0.0071 | 0.9142±0.0078 |
| Drug cold start | 0.9164±0.0093 | 0.9125±0.0068 |
| Target cold start | 0.8388±0.0272 | 0.8335±0.0258 |

Table 10: Model performance comparison on activation dataset in terms of AUPR when using different negative sample strategies: Cross-Negative vs Current-Negative (Mean±Std).

| Split Type | Cross-Negative | Current-Negative |
|---|---|---|
| Warm start | 0.6326±0.0232 | 0.6424±0.0221 |
| Drug cold start | 0.6239±0.0245 | 0.6278±0.0222 |
| Target cold start | 0.4618±0.0313 | 0.4497±0.0374 |

### C.7 SENSITIVITY TO BATCH SIZE AND IN-BATCH NEGATIVES

Our method uses the standard in-batch negative formulation: for a batch of size $N$, each sample uses the other $N - 1$ samples as negatives. We do not use stabilization techniques such as memory banks. To test sensitivity, we varied the per-GPU batch size from 32 up to 512. The main paper results used a batch size of 1280. As expected, performance generally improves with larger batch sizes, since more in-batch negatives benefit the contrastive and volume losses. The corresponding results are shown in table 11 and 12.

Table 11: AUROC Performance vs. Batch Size (Activation Dataset)

| Batch Size | Warm Start | Drug Cold Start | Target Cold Start |
|---|---|---|---|
| 32 | 0.901 | 0.906 | 0.819 |
| 128 | 0.905 | 0.906 | 0.821 |
| 512 | 0.916 | 0.918 | 0.839 |
| 1280 | 0.914 | 0.913 | 0.834 |

Note that the 512 batch size results are very close to the 1280 results, suggesting performance may begin to saturate beyond batch size 512.

### C.8 DOWNSTREAM TASK ARCHITECTURE

For drug-target interaction (DTI) prediction evaluation, we employ a lightweight classification architecture that leverages the pre-trained embeddings from our four-modal framework. The downstream architecture is detailed in Algorithm 3 and uses only the drug (SMILES) and protein modalities relevant for binding prediction.

### C.9 EVALUATION METRICS

We employ five standard binary classification metrics to comprehensively assess DTI prediction performance. Given the confusion matrix with true positives (TP), false positives (FP), true negatives (TN), and false negatives (FN), the metrics are defined as follows:

Table 12: AUPRC Performance vs. Batch Size (Activation Dataset)

| Batch Size | Warm Start | Drug Cold Start | Target Cold Start |
|---|---|---|---|
| 32 | 0.615 | 0.607 | 0.432 |
| 128 | 0.625 | 0.619 | 0.438 |
| 512 | 0.642 | 0.629 | 0.458 |
| 1280 | 0.642 | 0.628 | 0.450 |

---

**Algorithm 3** Drug-Target Interaction Prediction

---

**Require:** Pre-trained embeddings $f^s, f^p \in \mathbb{R}^{512}$
**Require:** Drug-protein pair $(x_i^s, x_j^p)$, binding label $y_{ij} \in \{0, 1\}$
**Ensure:** Binding prediction $\hat{y}_{ij}$
  1: $f_i^s \leftarrow \text{FROZEN}(F_\phi^s(E_s(x_i^s)))$ {Use pre-trained SMILES embedding}
  2: $f_j^p \leftarrow \text{FROZEN}(F_\phi^p(E_p(x_j^p)))$ {Use pre-trained protein embedding}
  3: $f^{\text{concat}} \leftarrow [f_i^s; f_j^p] \in \mathbb{R}^{1024}$ {Concatenate embeddings}
  4: $h_1 \leftarrow \text{ReLU}(\text{Linear}_{1024 \to 512}(f^{\text{concat}}))$
  5: $h_1 \leftarrow \text{Dropout}_{0.3}(h_1)$
  6: $h_2 \leftarrow \text{ReLU}(\text{Linear}_{512 \to 256}(h_1))$
  7: $h_2 \leftarrow \text{Dropout}_{0.3}(h_2)$
  8: $\text{logits} \leftarrow \text{Linear}_{256 \to 2}(h_2)$
  9: $\hat{y}_{ij} \leftarrow \arg\max(\text{softmax}(\text{logits}))$
10: **return** $\hat{y}_{ij}$

---

**Area Under ROC Curve (AUROC)** AUROC measures the model's ability to discriminate between positive and negative classes across all classification thresholds:

$$\text{AUROC} = \int_0^1 \text{TPR}(\text{FPR}^{-1}(t)) \, dt \tag{9}$$

where $\text{TPR} = \frac{\text{TP}}{\text{TP+FN}}$ and $\text{FPR} = \frac{\text{FP}}{\text{FP+TN}}$.

**Area Under Precision-Recall Curve (AUPRC)** AUPRC is particularly informative for imbalanced datasets and measures performance across different precision-recall trade-offs:

$$\text{AUPRC} = \int_0^1 \text{Precision}(\text{Recall}^{-1}(t)) \, dt \tag{10}$$

where $\text{Precision} = \frac{\text{TP}}{\text{TP+FP}}$ and $\text{Recall} = \frac{\text{TP}}{\text{TP+FN}}$.

**Sensitivity (Recall)** Sensitivity measures the proportion of actual positive cases correctly identified:

$$\text{Sensitivity} = \frac{\text{TP}}{\text{TP} + \text{FN}} \tag{11}$$

**F1-Score** F1-score provides the harmonic mean of precision and recall, balancing both measures:

$$\text{F1} = 2 \cdot \frac{\text{Precision} \times \text{Recall}}{\text{Precision} + \text{Recall}} = \frac{2 \cdot \text{TP}}{2 \cdot \text{TP} + \text{FP} + \text{FN}} \tag{12}$$

**Accuracy** Accuracy measures the overall proportion of correct predictions:

$$\text{Accuracy} = \frac{\text{TP} + \text{TN}}{\text{TP} + \text{TN} + \text{FP} + \text{FN}} \tag{13}$$

# D ABLATION STUDY

To complement the ablation study presented in Section 4.5 on the Activation dataset, we provide additional comprehensive ablation experiments on the Yamanishi 08 dataset in Figure 5. This ad-

ditional evaluation allows us to assess the generalizability of our component contributions across different datasets and task characteristics.

## D.1 EXPERIMENTAL SETUP

The ablation study on Yamanishi 08 follows the same experimental configuration as described in Section 4.5, evaluating five distinct setups:

- **Exp 1**: Full GRAM-DTI model with all components and adaptive modality dropout
- **Exp 2**: Training without Gramian volume-based loss ($L = \lambda_2 L_{\text{bi}} + \lambda_3 L_{\text{IC50}}$)
- **Exp 3**: Training without bimodal contrastive loss ($L = \lambda_1 L_{\text{vol}} + \lambda_3 L_{\text{IC50}}$)
- **Exp 4**: Training without IC50 auxiliary supervision ($L = \lambda_1 L_{\text{vol}} + \lambda_2 L_{\text{bi}}$)
- **Exp 5**: Training with full objective but without adaptive modality dropout

## D.2 RESULTS ANALYSIS

The results on Yamanishi 08, shown in Figure 5, demonstrate consistent patterns with those observed on the Activation dataset, confirming the robustness of our design choices across different datasets.

**Consistent Superior Performance of Full Model**: Across all three data splitting scenarios (warm start, drug cold start, target cold start) and five evaluation metrics (AUROC, AUPRC, Sensitivity, F1, Accuracy), the full GRAM-DTI model (Exp 1) generally achieves the highest performance, demonstrating the synergistic benefit of all proposed components.

1. The volume-based multimodal alignment provides substantial benefits over traditional pairwise approaches
2. Adaptive modality dropout prevents overfitting and improves generalization
3. IC50 auxiliary supervision enhances biological relevance of learned representations
4. The synergistic combination of all components yields optimal performance

These consistent findings across different datasets and evaluation scenarios validate the generalizability of our GRAM-DTI framework design principles.

# E   ADDITIONAL EXPERIMENTAL DETAILS

Table 13: Performance metrics with standard deviations for GRAM-DTI across all evaluation datasets and data splitting scenarios. Results are reported as mean ± standard deviation across cross-validation folds.

| Dataset | Split Type | AUROC ↑ | AUPRC↑ | Sensitivity↑ | F1 ↑ | Accuracy ↑ |
|---|---|---|---|---|---|---|
| Yamanishi_08 | warm start | 0.9771±0.0042 | 0.9036±0.0079 | 0.7954±0.0152 | 0.8353±0.0096 | 0.9715±0.0015 |
| | drug cold start | 0.8279±0.0285 | 0.4404±0.0662 | 0.2020±0.0575 | 0.3090±0.0693 | 0.9193±0.0134 |
| | target cold start | 0.9553±0.0155 | 0.8494±0.0312 | 0.7189±0.0453 | 0.7840±0.0285 | 0.9643±0.0042 |
| Hetionet | warm start | 0.9808±0.0011 | 0.8586±0.0082 | 0.7580±0.0085 | 0.7891±0.0065 | 0.9632±0.0010 |
| | drug cold start | 0.8550±0.0385 | 0.5291±0.0626 | 0.2981±0.0645 | 0.4227±0.0619 | 0.9273±0.0131 |
| | target cold start | 0.9210±0.0079 | 0.6258±0.0239 | 0.4569±0.0448 | 0.5502±0.0319 | 0.9325±0.0038 |
| Activation | warm start | 0.9142±0.0078 | 0.6424±0.0221 | 0.5155±0.0240 | 0.5950±0.0075 | 0.9364±0.0026 |
| | drug cold start | 0.9125±0.0068 | 0.6278±0.0222 | 0.5135±0.0349 | 0.5879±0.0186 | 0.9347±0.0030 |
| | target cold start | 0.8335±0.0258 | 0.4497±0.0374 | 0.2451±0.0591 | 0.3447±0.0620 | 0.9168±0.0104 |
| Inhibition | warm start | 0.9491±0.0018 | 0.7849±0.0061 | 0.6588±0.0109 | 0.7202±0.0061 | 0.9535±0.0013 |
| | drug cold start | 0.9398±0.0018 | 0.7555±0.0034 | 0.5949±0.0176 | 0.6801±0.0081 | 0.9492±0.0011 |
| | target cold start | 0.8234±0.0218 | 0.4641±0.0559 | 0.2584±0.0827 | 0.3687±0.0872 | 0.9220±0.0087 |

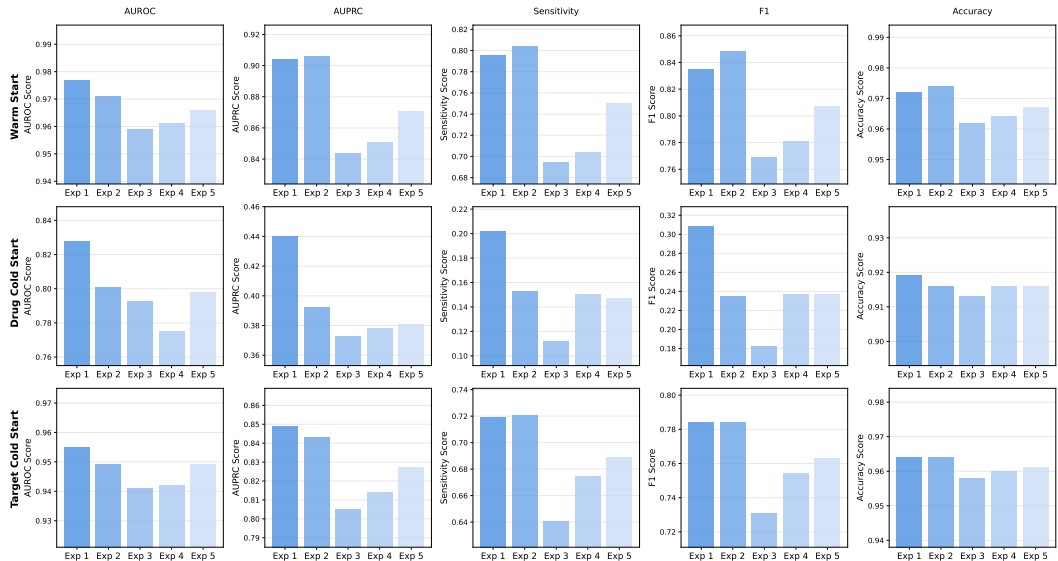

Figure 5: Ablation study results on the Yamanishi 08 dataset across five experimental configurations and three data splitting scenarios. The full GRAM-DTI model (Exp 1) consistently outperforms variants with removed components across most metrics and scenarios, demonstrating the robust contribution of each training objective component. Results complement those shown in Figure 2 (Activation dataset) and confirm the generalizability of our design choices across different DTI prediction benchmarks.

### E.1 STANDARD DEVIATION RESULTS FOR MAIN PERFORMANCE COMPARISON

Table 13 provides comprehensive performance statistics for GRAM-DTI, including standard deviations across all evaluation metrics, datasets, and data splitting scenarios. These detailed statistics demonstrate the stability and reliability of our approach across cross-validation folds.

### E.2 PERFORMANCE ANALYSIS WHEN CERTAIN MODALITIES ARE MISSING AT PRETRAINING TIME

Bringing in as many relevant modalities in the pretraining would help learn better representation for the corresponding downstream task. However, what if during training, not all the modalities are available? We investigated this question by considering scenario where certain modality is not available during pretraining. For drug-target interaction tasks, the drug and protein modalities are crucial. If either is unavailable during pretraining, the model cannot learn joint embeddings for interacting pairs. if for one the missing modality, what one can do is use only the embedding obtained from original encoders (ESM2/molformer) without further fine tuning it with the contrastive learning, this scenario will effectively falls back to a regime similar to DTIAM, where drug and protein embeddings are learned separately. This highlights a key strength of GRAM-DTI: by pretraining with both modalities present, it learns aligned embeddings that improve downstream performance. For auxiliary modalities such as functional descriptors and HTA:

- If both are missing during pretraining, this is equivalent to the ablation study (Exp2), where the volume-loss component is removed (as those two modality anticipate in the training through volume loss). In this case, we observe a slight decrease in performance, indicating that these modalities provide useful signals for alignment.

- If only one is missing, as shown in Table 14 and 15 the model still benefits from the available modality, with a moderate drop in performance. This demonstrates that GRAM-DTI can gracefully handle partial modality availability, but full multi-modal pretraining yields the strongest embeddings.

Table 14: Model performance on the Activation dataset in terms of $AUROC$ for different modality configurations: 3-mod-no-text: all modality except functional descriptors are available during pretraining, 3-mod-no-HTA: all modality except HTA are available during pretraining , 4-mod: all four modalities are available during pretraining (current setup).

| Split Type | 3-mod-no-text | 3-mod-no-HTA | 4-mod |
|---|---|---|---|
| Warm start | 0.907 | 0.901 | 0.914 |
| Drug cold start | 0.907 | 0.903 | 0.913 |
| Target cold start | 0.828 | 0.821 | 0.834 |

Table 15: Model performance on the Activation dataset in terms of $AUPRC$ for different modality configurations: 3-mod-no-text: all modality except functional descriptors are available during pretraining, 3-mod-no-HTA: all modality except HTA are available during pretraining , 4-mod: all four modalities are available during pretraining (current setup).

| Split Type | 3-mod-no-text | 3-mod-no-HTA | 4-mod |
|---|---|---|---|
| Warm start | 0.609 | 0.606 | 0.642 |
| Drug cold start | 0.615 | 0.611 | 0.628 |
| Target cold start | 0.437 | 0.440 | 0.450 |

### E.3 OVERLAP OF ENTITIES ANALYSIS BETWEEN PRETRAINING AND DOWNSTREAM TASK

To verify that our method does not memorize entity-specific patterns, we conducted an overlap analysis on the Activation dataset between pretraining and downstream task dataset (Other datasets are shown in Table 17). The results revealed 236 overlapping proteins and 314 overlapping SMILES. We removed all pairs containing these overlapping entities from the pretraining data, resulting in 6,065 exact (SMILES, protein) pairs removed (11.9% of pretraining data). The results are shown in Table 16. Despite removing nearly 12% of the pretraining data, the performance drops are modest

Table 16: Performance before and after cleaning on the Activation dataset.

| Split Type | Metric | Before Cleaning | After Cleaning | $\Delta$ |
|---|---|---|---|---|
| Warm Start | $AUROC$ | 0.914 | 0.901 | -0.013 |
| | $AUPRC$ | 0.642 | 0.613 | -0.029 |
| Drug Cold Start | $AUROC$ | 0.913 | 0.905 | -0.008 |
| | $AUPRC$ | 0.628 | 0.624 | -0.004 |
| Target Cold Start | $AUROC$ | 0.834 | 0.795 | -0.039 |
| | $AUPRC$ | 0.450 | 0.389 | -0.061 |

across all splits (0.01-0.03). This demonstrates that our model's strong performance is not primarily driven by memorizing exact pairs, and further validates our cold-start claims:

- Entity overlap contributes to performance but is not the dominant factor
- The 4-modal learning framework captures transferable molecular representations rather than memorizing specific entity combinations
- Drug cold-start generalization is particularly robust ($\Delta$ AUROC = -0.008), showing minimal sensitivity to entity overlap

### E.4 MODEL PERFORMANCE SENSITIVITY TO THE NEGATIVE-SAMPLE RATIO IN THE DOWNSTREAM TASK

All baselines generate negative samples at a 1:10 ratio relative to positive samples across datasets; to ensure a fair comparison, we adopt the same setup. To evaluate sensitivity to this choice, we also report results using alternative negative-sample ratios. In table 18, we show our model performance as well as best baseline (DTIAM) on the Activation dataset when negative samples are generated at various ratio with respect to positive samples.

Table 17: Entity-level overlap analysis between pretraining and downstream datasets. "Pairs Sharing Any Entity" denotes the number of pretraining pairs containing at least one protein or SMILES that appears in the downstream dataset. Note that exact interacting (SMILES, Protein) pairs present in the downstream tasks have already been strictly removed from the pretraining set to prevent data leakage.

| Dataset | Total Pairs | Proteins | SMILES | Pairs Sharing Any Entity | Percentage |
|---|---|---|---|---|---|
| activation | 50,968 | 236 | 314 | 6,065 | 11.90% |
| Hetionet | 50,968 | 1,936 | 853 | 42,242 | 82.88% |
| inhibition | 50,968 | 860 | 1,382 | 36,372 | 71.36% |
| yamanishi_08 | 50,968 | 556 | 344 | 20,223 | 39.68% |

Table 18: Performance Comparison of GRAM-DTI and Baseline on Activation Dataset under Different Ratios

| Ratio | Split Type | Method | AUROC | AUPRC | Precision | Recall |
|---|---|---|---|---|---|---|
| 1:1 | Warm Start | GRAM-DTI | **0.862** | **0.850** | **0.813** | **0.835** |
| | | DTIAM | 0.855 | 0.841 | 0.764 | 0.822 |
| | Drug Cold Start | GRAM-DTI | **0.853** | **0.843** | **0.822** | 0.818 |
| | | DTIAM | 0.831 | 0.829 | 0.794 | **0.882** |
| | Target Cold Start | GRAM-DTI | **0.790** | **0.784** | **0.788** | **0.691** |
| | | DTIAM | 0.740 | 0.737 | 0.705 | 0.656 |
| 1:5 | Warm Start | GRAM-DTI | **0.893** | **0.688** | **0.751** | **0.634** |
| | | DTIAM | 0.888 | 0.674 | 0.704 | 0.609 |
| | Drug Cold Start | GRAM-DTI | **0.888** | **0.674** | **0.742** | 0.616 |
| | | DTIAM | 0.870 | 0.654 | 0.739 | **0.712** |
| | Target Cold Start | GRAM-DTI | **0.818** | **0.559** | **0.680** | **0.412** |
| | | DTIAM | 0.792 | 0.509 | 0.566 | 0.365 |
| 1:10 | Warm Start | GRAM-DTI | **0.914** | **0.642** | **0.697** | 0.502 |
| | | DTIAM | 0.903 | 0.623 | 0.670 | **0.540** |
| | Drug Cold Start | GRAM-DTI | **0.913** | **0.628** | 0.692 | 0.491 |
| | | DTIAM | 0.907 | 0.611 | **0.725** | **0.583** |
| | Target Cold Start | GRAM-DTI | **0.834** | **0.450** | **0.638** | 0.260 |
| | | DTIAM | 0.792 | 0.391 | 0.533 | **0.293** |

## E.5 HANDLING PARTIAL-MODALITY DATA DURING PRETRAINING

we can extend pretraining to include samples with missing modalities, which would substantially increase the size of our training set. To assess whether all modalities are beneficial, our current pretraining phase includes only samples in which all four modalities are present, a choice that significantly limits the dataset. As a proof of concept, we evaluated whether including samples with only a subset of modalities improves downstream performance. From the pretraining dataset we created:

- Fully Observed (80%): 80% of the original data, kept unchanged.

- Partially Observed (20%): the remaining 20% where we randomly dropped one modality.

We compared training on only the 80% fully-observed subset vs. training on the full 100% dataset (80% full + 20% partial) using a masked-volume loss for the partial samples. This simulates shows how pretraining can be expanded when some modalities are missing at random.

As the table 19 shows, incorporating the 20% partial data via masked-volume training improves performance across all metrics and splits, with a notable improvement on the Target Cold Start (AUROC 0.828 vs. 0.791).

Table 19: Performance with Partial-Modality ("Masked-Volume") Training

| Split Type | Metric | Fully Observed Only (80% data) | Full + Partial (100% data) |
|---|---|---|---|
| Warm Start | AUROC | 0.905 | 0.912 |
| | AUPRC | 0.627 | 0.634 |
| Drug Cold Start | AUROC | 0.903 | 0.907 |
| | AUPRC | 0.613 | 0.615 |
| Target Cold Start | AUROC | 0.791 | 0.828 |
| | AUPRC | 0.422 | 0.437 |

### E.6 ANALYSIS OF DROPOUT VS. WEIGHTING STRATEGIES

To validate our Gradient-Informed Modality Dropout strategy, we compared it against two alternative "soft" balancing mechanisms on the Activation dataset:

- **Weighted-Modality Gradients:** Instead of dropping a modality, we scale its gradient by the inverse of its norm with probability $p_{drop}$.
- **Standard Weighted Loss:** We assign learnable weights to each modality's loss term to balance contributions without dropout.

As shown in Table 20, our probabilistic dropout strategy achieves the best performance. We hypothesize that probabilistically removing modalities forces the model to find alternative distinct paths for reasoning in the joint embedding space, acting as a stronger regularizer than soft weighting.

Table 20: Comparison of Modality Balancing Strategies on the Activation Dataset. Our hard dropout strategy outperforms soft weighting approaches.

| Strategy | Split Type | AUROC | AUPRC |
|---|---|---|---|
| Gradient-Informed Dropout (Ours) | Warm Start | **0.914** | **0.642** |
| | Drug Cold Start | **0.913** | **0.628** |
| | Target Cold Start | **0.834** | **0.450** |
| Weighted Gradients | Warm Start | 0.909 | 0.618 |
| | Drug Cold Start | 0.910 | 0.624 |
| | Protein Cold Start | 0.828 | 0.445 |
| Standard Weighted Loss | Warm Start | 0.901 | 0.621 |
| | Drug Cold Start | 0.892 | 0.619 |
| | Target Cold Start | 0.814 | 0.440 |

### E.7 EXPERIMENTS WITH STRONGER MOLECULAR ENCODERS

Our default GRAM-DTI implementation uses MolFormer for computational efficiency. To demonstrate the framework's extensibility, we replaced MolFormer with two larger, more advanced encoders: Uni-Mol2 (84M parameters) and BioT5+. As shown in Table 21, utilizing stronger encoders consistently improves performance across all splits, particularly in the challenging target Cold start scenario. It shows that stronger encoders yield consistently better results.

### E.8 FALSE NEGATIVE CASE ANALYSIS

To understand where the model fails, we systematically identified the top "hardest" false negatives in the Activation dataset—pairs where the model predicted a strong negative signal despite a positive ground truth label. These are listed in table 22 below:

### E.9 MOA TASK ADDITIONAL BASELINES

To extend the set of baselines beyond those used in the DTIAM study for the MoA task, we included two additional methods: DeepDTA (Öztürk et al., 2018) and GraphDTA (Nguyen et al., 2021).

Table 21: Sensitivity analysis using advanced molecular encoders on the Activation dataset

| Encoder | Split Type | AUROC | AUPRC |
|---|---|---|---|
| MolFormer (Original) | Warm Start | 0.9142 | 0.6424 |
| | Drug Cold Start | 0.9125 | 0.6278 |
| | Target Cold Start | 0.8335 | 0.4497 |
| Uni-Mol2 (84M) | Warm Start | **0.9280** | 0.6768 |
| | Drug Cold Start | **0.9270** | 0.6658 |
| | Target Cold Start | **0.8642** | **0.4848** |
| BioT5+ | Warm Start | 0.9273 | **0.6828** |
| | Drug Cold Start | 0.9254 | **0.6840** |
| | Target Cold Start | 0.8577 | 0.4805 |

Table 22: Top-10 False Negative Pairs (Drug ID & Target ID)

| Rank | Drug ID | Target ID |
|---|---|---|
| 1 | D0NY1R | T36075 |
| 2 | D08FKH | T12475 |
| 3 | D0G2VT | T59604 |
| 4 | D0JB3H | T88505 |
| 5 | D0L5WA | T28893 |
| 6 | D0K8NR | T72458 |
| 7 | D03LQC | T52522 |
| 8 | D03XIS | T92076 |
| 9 | D07QAK | T28893 |
| 10 | D0AJ2T | T88505 |

Table 23: Performance comparison between GRAM-DTI and state-of-the-art baselines (DeepDTA, GraphDTA, AI-DTI, DTIAM) on MoA prediction tasks across multiple datasets and data splitting scenario. GRAM-DTI demonstrates superior performance in most evaluation settings. † indicates reproduced results; other results are from baseline papers. **Bold** denotes best performance.

| Data | Metric | Scenario | DeepDTA† | GraphDTA† | AI-DTI | DTIAM† | GRAM-DTI |
|---|---|---|---|---|---|---|---|
| Activation | AUPR | Warm Start | 0.246±0.0232 | 0.282±0.0240 | 0.583 | 0.623±0.0245 | **0.642±0.0221** |
| | | Drug Cold Start | 0.255±0.0209 | 0.298±0.0195 | 0.550 | 0.611±0.0252 | **0.628±0.0222** |
| | | Cold Start | 0.109±0.0163 | 0.124±0.0175 | 0.219 | 0.391±0.0320 | **0.450±0.0374** |
| | AUROC | Warm Start | 0.759±0.0200 | 0.784±0.0185 | 0.888 | 0.903±0.0088 | **0.914±0.0078** |
| | | Drug Cold Start | 0.765±0.0059 | 0.796±0.0062 | 0.879 | 0.907±0.0076 | **0.913±0.0068** |
| | | Cold Start | 0.573±0.0241 | 0.588±0.0255 | 0.652 | 0.792±0.0240 | **0.834±0.0258** |
| Inhibition | AUPR | Warm Start | 0.542±0.0195 | 0.585±0.0280 | 0.840 | **0.845±0.0070** | 0.785±0.0061 |
| | | Drug Cold Start | 0.531±0.0170 | 0.592±0.0195 | **0.830** | 0.731±0.0045 | 0.756±0.0034 |
| | | Cold Start | 0.265±0.0210 | 0.284±0.0312 | 0.215 | 0.445±0.0620 | **0.464±0.0559** |
| | AUROC | Warm Start | 0.854±0.0105 | 0.872±0.0098 | 0.952 | **0.954±0.0025** | 0.949±0.0018 |
| | | Drug Cold Start | 0.849±0.0185 | 0.876±0.0115 | **0.948** | 0.921±0.0028 | 0.940±0.0018 |
| | | Cold Start | 0.635±0.0220 | 0.649±0.0117 | 0.605 | 0.819±0.0205 | **0.823±0.0028** |

Although DTIAM remains the strongest baseline overall, our model GRAM-DTI achieves superior performance in most evaluation settings. The results are shown in the table 23.

### E.10 SIGNIFICANCE TEST

We ran one-sided Welch t-tests to assess whether GRAM-DTI outperforms DTIAM (the strongest baseline). Tests were computed from summary statistics (means and standard deviations) using n=10 (10 folds) for Yamanishi_08 and Hetionet dataset and n=5 (5 folds) for Activation and Inhibition dataset with the one-sided hypothesis H1: GRAM-DTI > DTIAM. Table 24 and Table 25 report

the corresponding p values for the MoA and DTI tasks, respectively, and Tables 26 and 27 show the zero-shot retrieval results. Cells highlighted in light blue indicate the better method in each row. Note that we performed a total of 48 tests. To control for multiple comparisons we applied a Bonferroni correction and used an adjusted significance threshold of: $p^{adjusted} = \frac{0.05}{48} \approx 0.00104$, rather than the conventional $p = 0.05$. p-values highlighted in light blue indicate $p < 0.00104$.

Table 24: Significance test on the MoA task: means ($\pm$ std) and one-sided $p$-values for $H_1$: GRAM > DTIAM (independent Welch test). One-sided $p$-values with $p < 0.00104$ are highlighted in light blue.

| Dataset | Metric | Scenario | DTIAM ($\mu \pm \sigma$) | GRAM ($\mu \pm \sigma$) | one-sided $p$ value |
|---|---|---|---|---|---|
| **Activation** | AUPR | Warm start | $0.623 \pm 0.0245$ | $0.642 \pm 0.0221$ | 0.1171 |
| | | Drug cold start | $0.611 \pm 0.0252$ | $0.628 \pm 0.0222$ | 0.1455 |
| | | Target cold start | $0.391 \pm 0.0320$ | $0.450 \pm 0.0374$ | 0.0143 |
| | AUROC | Warm start | $0.903 \pm 0.0088$ | $0.914 \pm 0.0078$ | 0.0352 |
| | | Drug cold start | $0.907 \pm 0.0076$ | $0.913 \pm 0.0068$ | 0.1126 |
| | | Target cold start | $0.792 \pm 0.0240$ | $0.834 \pm 0.0258$ | 0.0144 |
| **Inhibition** | AUPR | Warm start | $0.845 \pm 0.0070$ | $0.785 \pm 0.0061$ | 1.0000 |
| | | Drug cold start | $0.731 \pm 0.0045$ | $0.756 \pm 0.0034$ | < 0.0001 |
| | | Target cold start | $0.445 \pm 0.0620$ | $0.464 \pm 0.0559$ | 0.3123 |
| | AUROC | Warm start | $0.954 \pm 0.0025$ | $0.949 \pm 0.0018$ | 0.9961 |
| | | Drug cold start | $0.921 \pm 0.0028$ | $0.940 \pm 0.0018$ | < 0.0001 |
| | | Target cold start | $0.819 \pm 0.0205$ | $0.823 \pm 0.0028$ | 0.3435 |

Table 25: Significance test on the DTA task: means ($\pm$ std) and one-sided $p$-values for $H_1$: GRAM > DTIAM (independent Welch test). One-sided $p$-values with $p < 0.00104$ are highlighted in light blue.

| Dataset | Metric | Scenario | DTIAM ($\mu \pm \sigma$) | GRAM ($\mu \pm \sigma$) | one-sided $p$ value |
|---|---|---|---|---|---|
| **Yamanishi 08** | AUPR | Warm start | $0.901 \pm 0.0085$ | $0.904 \pm 0.0079$ | 0.2122 |
| | | Drug cold start | $0.439 \pm 0.0580$ | $0.440 \pm 0.0662$ | 0.4859 |
| | | Target cold start | $0.844 \pm 0.0350$ | $0.849 \pm 0.0312$ | 0.3700 |
| | AUROC | Warm start | $0.967 \pm 0.0050$ | $0.977 \pm 0.0042$ | < 0.0001 |
| | | Drug cold start | $0.818 \pm 0.0255$ | $0.828 \pm 0.0285$ | 0.2096 |
| | | Target cold start | $0.941 \pm 0.0180$ | $0.955 \pm 0.0155$ | 0.0395 |
| **Hetionet** | AUPR | Warm start | $0.879 \pm 0.0095$ | $0.859 \pm 0.0082$ | 0.9984 |
| | | Drug cold start | $0.514 \pm 0.0680$ | $0.529 \pm 0.0626$ | 0.3070 |
| | | Target cold start | $0.625 \pm 0.0210$ | $0.626 \pm 0.0239$ | 0.4610 |
| | AUROC | Warm start | $0.957 \pm 0.0015$ | $0.981 \pm 0.0011$ | < 0.0001 |
| | | Drug cold start | $0.752 \pm 0.0355$ | $0.855 \pm 0.0385$ | < 0.0001 |
| | | Target cold start | $0.917 \pm 0.0090$ | $0.921 \pm 0.0079$ | 0.1525 |

### E.11 ZERO-SHOT RETRIEVAL TASK METHODOLOGY

This section provides detailed methodology for the zero-shot retrieval experiments presented in Section 4.3 of the main text.

#### E.11.1 TASK FORMULATION

The zero-shot retrieval task evaluates GRAM-DTI's ability to identify relevant drug-target pairs using only the learned multimodal representations, without any task-specific fine-tuning. We formulate two complementary retrieval scenarios:

- **Drug-to-Protein Retrieval (S→P):** Given a query drug (SMILES representation), retrieve the most relevant target proteins from a candidate set.

Table 26: Significance test on Zero-shot retrieval task (Yamanishi_08 and Hetionet): one-sided Welch $t$-test ($H_1$: GRAM > DTIAM), values $p < 0.00104$ are highlighted.

| Direction | Metric | Yamanishi_08 | | | Hetionet | | |
|---|---|---|---|---|---|---|---|
| | | DTIAM | GRAM | $p$ | DTIAM | GRAM | $p$ |
| S→P | R@1 | $0.0038 \pm 0.0004$ | $\mathbf{0.0465 \pm 0.0027}$ | $< 0.0001$ | $0.0043 \pm 0.0002$ | $\mathbf{0.0331 \pm 0.0038}$ | $< 0.0001$ |
| | R@10 | $0.0341 \pm 0.0042$ | $\mathbf{0.1691 \pm 0.0084}$ | $< 0.0001$ | $0.0434 \pm 0.0051$ | $\mathbf{0.1340 \pm 0.0025}$ | $< 0.0001$ |
| | R@100 | $0.1960 \pm 0.0181$ | $\mathbf{0.4449 \pm 0.0075}$ | $< 0.0001$ | $0.2066 \pm 0.0109$ | $\mathbf{0.3616 \pm 0.0063}$ | $< 0.0001$ |
| P→S | R@1 | $0.0040 \pm 0.0002$ | $\mathbf{0.0742 \pm 0.0120}$ | $< 0.0001$ | $0.0404 \pm 0.0028$ | $0.0236 \pm 0.0010$ | $1.0000$ |
| | R@10 | $0.0849 \pm 0.0089$ | $\mathbf{0.2465 \pm 0.0256}$ | $< 0.0001$ | $0.1319 \pm 0.0095$ | $0.1049 \pm 0.0055$ | $1.0000$ |
| | R@100 | $0.3670 \pm 0.0186$ | $\mathbf{0.5540 \pm 0.0148}$ | $< 0.0001$ | $0.3632 \pm 0.0474$ | $0.3841 \pm 0.0082$ | $0.1900$ |

Table 27: Significance test on Zero-shot retrieval task (Activation and Inhibition): One-sided Welch $t$-test ($H_1$: GRAM > DTIAM), $p < 0.00104$ values are highlighted.

| Direction | Metric | Activation | | | Inhibition | | |
|---|---|---|---|---|---|---|---|
| | | DTIAM | GRAM | $p$ | DTIAM | GRAM | $p$ |
| S→P | R@1 | $0.0028 \pm 0.0002$ | $\mathbf{0.0136 \pm 0.0011}$ | $< 0.0001$ | $0.0004 \pm 0.0000$ | $\mathbf{0.0055 \pm 0.0003}$ | $< 0.0001$ |
| | R@10 | $0.0266 \pm 0.0037$ | $\mathbf{0.1020 \pm 0.0067}$ | $< 0.0001$ | $0.0097 \pm 0.0006$ | $\mathbf{0.0337 \pm 0.0011}$ | $< 0.0001$ |
| | R@100 | $0.3184 \pm 0.0229$ | $\mathbf{0.5688 \pm 0.0172}$ | $< 0.0001$ | $0.1036 \pm 0.0104$ | $\mathbf{0.1994 \pm 0.0018}$ | $< 0.0001$ |
| P→S | R@1 | $0.0071 \pm 0.0008$ | $\mathbf{0.0370 \pm 0.0069}$ | $< 0.0001$ | $0.0000 \pm 0.0000$ | $\mathbf{0.0221 \pm 0.0061}$ | $< 0.0001$ |
| | R@10 | $0.0463 \pm 0.0050$ | $\mathbf{0.2454 \pm 0.0142}$ | $< 0.0001$ | $0.0028 \pm 0.0004$ | $\mathbf{0.0819 \pm 0.0065}$ | $< 0.0001$ |
| | R@100 | $0.2206 \pm 0.0264$ | $\mathbf{0.6029 \pm 0.0231}$ | $< 0.0001$ | $0.0588 \pm 0.0049$ | $\mathbf{0.2325 \pm 0.0094}$ | $< 0.0001$ |

- **Protein-to-Drug Retrieval (P→S):** Given a query protein (sequence representation), retrieve the most relevant drugs from a candidate set.

### E.11.2   EXPERIMENTAL SETUP

For each dataset, we construct retrieval queries and candidate pools as follows:

**Query and Candidate Construction:**

- For each known drug-target interaction $(d_i, p_j)$ in the data set, we treat $d_i$ as a query and all proteins in the dataset as candidates for S→P retrieval
- Similarly, we treat $p_j$ as a query and all drugs as candidates for P→S retrieval
- Ground truth relevance is determined by known interactions in the original datasets

**Embedding Generation:** We generate embeddings using the pre-trained GRAM-DTI framework:

- SMILES sequences are encoded using MoLFormer-XL, producing 768-dimensional representations
- Protein sequences are encoded using ESM-2, producing 1280-dimensional representations
- Both modalities are projected to a shared 512-dimensional space using trained projectors from the multimodal pre-training phase
- All embeddings are L2-normalized for cosine similarity computation

**Similarity Computation:** We compute cosine similarity between query and candidate representations using the projected embeddings:

$$\text{sim}(q, c) = \frac{f_q \cdot f_c}{\|f_q\|\|f_c\|} \tag{14}$$

where $f_q$ and $f_c$ are the normalized projected embeddings for query $q$ and candidate $c$, respectively.

**Ranking and Evaluation:**

1. For each query, we rank all candidates by their similarity scores in descending order
2. We evaluate retrieval performance using standard ranking metrics:
   - **Recall@1 (R@1):** Proportion of queries where the top-ranked candidate is relevant

- **Recall@10 (R@10):** Proportion of queries where at least one relevant item appears in the top-10 results
- **Recall@100 (R@100):** Proportion of queries where at least one relevant item appears in the top-100 results

### E.11.3   RETRIEVAL TASK ILLUSTRATION

Figure 6 illustrates the zero-shot retrieval evaluation process. Given a query protein $p_j$, the model computes cosine similarities with all candidate drugs in the dataset and ranks them by similarity scores. Retrieval metrics (R@1, R@10, R@100) measure whether known positive drug-target interactions appear within the top-k ranked candidates.

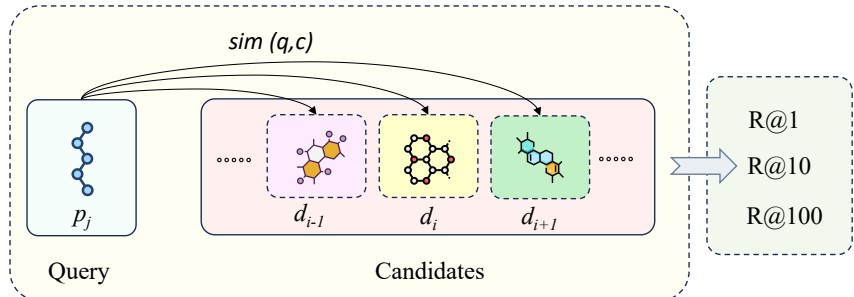

Figure 6: Illustration of zero-shot retrieval evaluation. A query protein $p_j$ is compared against all candidate drugs $\{d_{i-1}, d_i, d_{i+1}, ...\}$ using cosine similarity of learned embeddings. Recall@k metrics evaluate whether any known positive interactions appear in the top-k retrieved candidates.

### E.11.4   IMPLEMENTATION DETAILS

**Model Architecture:** We utilize the same encoder architectures and projector networks as in the main pre-training framework:

- SMILES projector: $768 \rightarrow 768 \rightarrow 512 \rightarrow 512$ (with GELU, LayerNorm, Dropout)
- Protein projector: $1280 \rightarrow 768 \rightarrow 512 \rightarrow 512$ (with GELU, LayerNorm, Dropout)

**Batch Processing:** Due to computational constraints, embeddings are generated in batches of 16 sequences to manage memory usage while maintaining efficiency.

**No Additional Training:** Critically, no additional training or fine-tuning is performed for the retrieval task. We use the representations learned during the multimodal pre-training phase directly, demonstrating the quality of the learned representations.

**Evaluation Protocol:** Following standard practice in information retrieval, we compute metrics across all queries in each dataset and report average performance. The evaluation uses only positive interactions from the retrieval datasets, ensuring fair assessment of the model's ability to identify true drug-target relationships.

The strong performance of GRAM-DTI in this zero-shot setting (Table 2 in main text) demonstrates that our volume-based multimodal alignment successfully learns semantically meaningful representations that capture drug-target relationships without task-specific supervision.

## F   COMPREHENSIVE MULTIMODAL EMBEDDING EVOLUTION ANALYSIS

This section provides a comprehensive analysis of how GRAM-DTI learns unified multimodal representations across different sample sizes and training epochs. We examine embedding evolution patterns to understand the dynamics of volume-based multimodal alignment and validate the effectiveness of our adaptive modality dropout mechanism.

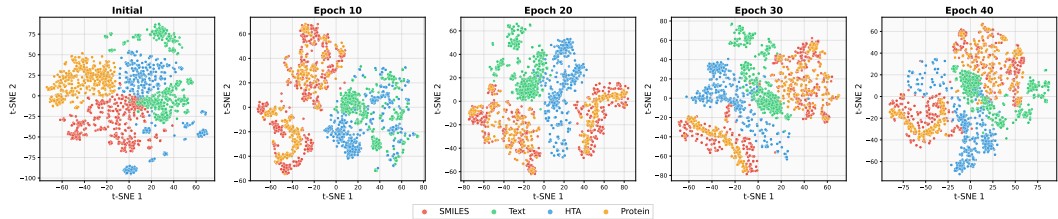

Figure 7: Embedding evolution analysis with 500 randomly sampled quadruplets, showing clear progression from separate modality clusters to integrated semantic representations.

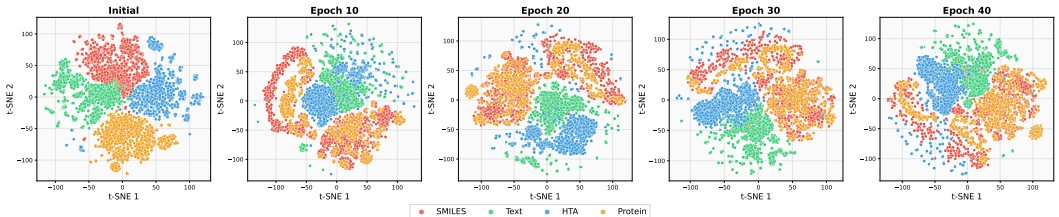

Figure 8: Embedding evolution analysis with 3,000 samples, demonstrating consistent patterns with reduced noise and clearer semantic sub-structures.

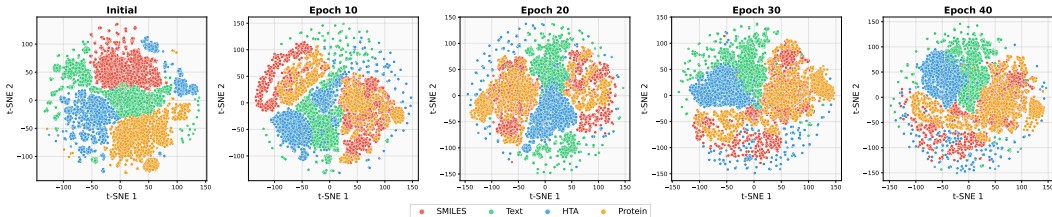

Figure 9: Embedding evolution analysis with 5,000 samples (shown in main text), providing optimal balance of detail and computational efficiency.

## F.1 EXPERIMENTAL SETUP

We conducted embedding evolution analysis across multiple scales to ensure robustness of our observations:

- **Sample sizes**: 500, 3,000, and 5,000 randomly selected quadruplets
- **Training epochs**: Initial state (epoch 0), 10, 20, 30, and 40
- **Visualization method**: t-SNE with perplexity=30, max_iter=1000
- **Preprocessing**: L2 normalization of projected embeddings, standardization per modality

For each epoch, we extracted embeddings from the four modalities using their respective pre-trained encoders (MolFormer-XL for SMILES, MolT5 for Text/HTA, ESM-2 for Protein), applied the trained projection layers to map into the unified 512-dimensional space, and performed t-SNE visualization.

## G LARGE LANGUAGE MODELS USAGE STATEMENT

We only used Large Language Models to correct grammars and polish the writing.

