# OpenReview forum: "GRAM-DTI: Adaptive Multimodal Representation Learning for Drug–Target Interaction Prediction"
_ICLR.cc/2026/Conference — ICLR 2026 Poster_

### Official Review · Reviewer_5mrg · 2025-10-23

**Soundness:** 3
**Presentation:** 2
**Contribution:** 2
**Rating:** 4
**Confidence:** 3

**Summary:**

This work introduces GRAM-DTI, a multimodal pretraining framework for Drug-Target Interaction. Firstly, GRAM-DTI combines multimodal modalities from molecule and protein into a unified representation using a novel volume-based contrastive learning to capture higher-order interdependencies when the modality number increases. Secondly, this paper proposes an adaptive modality dropout to dynamically filter out less informative modalities to prevent suboptimal representations. Thirdly, this work uses IC50 values as additional weak auxiliary supervision in the pretraining stage to make full use of the publicly available IC50 activity measurements. The GRAM-DTI achieves good performance on four datasets and several real-world application settings.

**Strengths:**

1. The motivations are interesting and attractive. Modality dropout could be a novel way to reduce the noise during the multimodal fusion, and the usage of new volume-based contrastive learning could capture more modality information when including more modalities.
2. The evaluation results show the performance improvement on four datasets, indicating the potential gain from the multimodal fusion. The ablation studies have been evaluated on most of the submodules, which is relatively comprehensive.

**Weaknesses:**

1. The choices of encoders are relatively out-of-date; for instance, there are stronger encoders for molecular modeling such as UniMol-v2, MLM-FG (2024), TranFoxMol (2023), MuMo (2025), ChemBERTa, HiGNN, 3D-MolT5, Mol-Llama, etc. A stronger encoder may bring better performance. The rationale of these selections is not specified.
2. For the modality dropout, the top and least contributed modalities have been dropped to pursue a balance in multimodal learning and prevent overfitting. However, how to make sure that the important information has been preserved and more noise has been filtered out? For example, if a modality creates a larger gradient than others, it may contain more important information in this specific case. In this situation, forcing the model to learn more from irrelevant modalities can even bring in more noise. Also, the larger gradient may tell the model more information about this input data. If you want to prevent overfitting, adding a weight factor may be a better approach rather than removing the whole modality.
3. The removal of modality losses (compared with weighted modality loss) in the modality drop approach may not be a meticulous design. It may cause the gradient to become unstable and harm the pretraining efficiency and even the performance.

**Questions:**

1. Is the encoder for molecule $E_m$​ trainable or not? In the pretraining stage of Figure 1, it did not show whether the $E_m$​ ​is trainable or not.
2. Why does this model use these relatively old encoders instead of modern ones? What's the reason for choosing these? Is there any ablation analysis to prove the selection?
3. What are the four F in the pretraining stage in Figure 1? What model structure is used for these modules?
4. For the datasets Inhibition and Activation, why only use AI-DTI and DTIAM baselines, and not stay consistent with the other two datasets?

---

> ### Author Response · Authors · 2025-11-19
> **Author response to review (and invitation for further discussion)- Response 1 of 2**
>
> We sincerely thank the reviewer for the constructive and thoughtful feedback. We also appreciate the recognition of our novel methodological contributions, the method’s strong performance, and comprehensive experimental evaluations.
>
> We are also thankful for the reviewer’s careful questions which we address point-by-point below. We have also included all experiments in the revised version of the paper and marked them in blue.
>
> #### **Weakness 1/Question 2: Experiments with stronger molecular encoders**: putational efficiency and we used commonly used encoders with stable and well-understood embeddings, which allowed us to focus on evaluating the effect of multi-modal alignment and modality dropout contributions primarily. Given the reviewer's suggestion, we have now run experiments with two stronger molecule encoders, namely UniMOl2 (84M) and BioT5+. Indeed, we observe that stronger encoder enhance the performance even further thus reinforcing the utility and modularity of our proposed framework which is seamlessly suitable for integrating better and stronger encoders if available in future. We have also added this analysis in Appendix Section E.7, and referred it in main text in lines 472-476.
>
> The results are summarized in the tables below:
>
> **Table 1: Performance with Advanced Encoders (Activation Dataset)**
> | Encoder | Split Type | AUROC (Mean) | AUPRC (Mean) |
> | :--- | :--- | :---: | :---: |
> | **MolFormer (Original)** | Warm Start | 0.9142 | 0.6424 |
> | | Drug Cold Start | 0.9125 | 0.6278 |
> | | Protein Cold Start | 0.8335 | 0.4497 |
> | **Uni-Mol2 (84M)** | Warm Start | **0.9280** | 0.6768 |
> | | Drug Cold Start | **0.9270** | 0.6658 |
> | | Protein Cold Start | **0.8642** | **0.4848** |
> | **BioT5+** | Warm Start | 0.9273 | **0.6828** |
> | | Drug Cold Start | 0.9254 | **0.6840** |
> | | Protein Cold Start | 0.8577 | 0.4805 |
>
> #### **Weakness 2/Weakness 3: Questions on whether modality dropout removes important information and potentially introduces gradient instability**: This is a very thoughtful question regarding the dynamics of multi-modal learning. We would like to clarify that our Gradient-Informed Modality Dropout is **probabilistically modulated strategy**. A modality meets the criteria for dropping (e.g., dominance) is not removed 100% of the time; it is only dropped with a probability $p_{drop}$ (tuned in our experiments). With probability $1-p_{drop}$, even a dominant modality is retained. This probabilistic design prevents over-reliance on a single modality (overfitting) while ensuring that strong signals are not permanently discarded.
>
> To validate our design choice against the reviewer's suggestion of "weighting" instead of "dropping," we implemented and tested two alternative strategies on the Activation dataset:
> 1.  **Weighted-Modality Gradients:** Instead of dropping, we scale the gradient of the targeted modality by $1/norm(gradient)$ with probability $p_{drop}$.
> 2.  **Standard Weighted Loss:** Assigning learnable weights to each modality's loss term to balance contributions, without any dropout.
>
> **Table 2: Comparison of Modality Balancing Strategies (Activation Dataset)**
> | Strategy | Split Type | AUROC (Mean) | AUPRC (Mean) |
> | :--- | :--- | :---: | :---: |
> | **Gradient-Informed Dropout (Ours)** | Warm Start | **0.914** | **0.642** |
> | | Drug Cold Start | **0.913** | **0.628** |
> | | Protein Cold Start | **0.834** | **0.450** |
> | **Weighted Gradients** | Warm Start | 0.909 | 0.618 |
> | | Drug Cold Start | 0.910 | 0.624 |
> | | Protein Cold Start | 0.828 | 0.445 |
> | **Standard Weighted Loss** | Warm Start | 0.901 | 0.621 |
> | | Drug Cold Start | 0.892 | 0.619 |
> | | Protein Cold Start | 0.814 | 0.440 |
> As shown, our Gradient-Informed Dropout strategy achieves the best trade-off. We hypothesize that "hard" dropout (applied probabilistically) forces the model to find alternative distinct paths for reasoning in the joint embedding space, essentially acting as a stronger regularizer than "soft" weighting, which might still allow the model to lazily rely on the dominant modality.  We have also added this analysis in Appendix Section E.6, and referred it in main text in lines 468-470.
>
> #### **Question 1: Is the encoder for molecule E_m ​ trainable or not? In the pretraining stage of Figure 1, it did not show whether the E_m​ ​is trainable or not.**: Thank you for noticing the accidental mistake in Figure 1! The molecule encoder E_m is frozen in our setup. We accidentally missed the fozen icon on E_m in Fig. 1 but explained it in main text lines 159-160. We have corrected this error and updated Fig. 1 showing E_m being frozen.

---

> > ### Author Response · Authors · 2025-11-19
> > **Author response to review (and invitation for further discussion)- Response 2 of 2**
> >
> > #### **Question 3: What are the four F in the pretraining stage in Figure 1? What model structure is used for these modules?**:
> >
> > The four F modules correspond to learnable projection layers that take embeddings from the frozen encoder and map them to a new space. These projections are trained through the loss function to produce aligned embeddings across all modalities. Each F module is implemented as a standard MLP layer. This has been explained in the paper in lines 159-160 in Appendix Section C.2.
> >
> > #### **Question 4: For the datasets Inhibition and Activation, why only use AI-DTI and DTIAM baselines, and not stay consistent with the other two datasets?**: We followed the exact evaluation protocol used in the DTIAM paper [1], which introduced the Activation and Inhibition datasets. In that work, the authors reported results only for AI-DTI and DTIAM, and did not provide comparisons with the broader set of baselines used for Yamanishi08 and Hetionet. This was not an omission on our side but a deliberate replication of prior experimental settings from prior work, consistent with the established benchmark.
> >
> > The reason why the DTIAM authors only use AI-DTI and DTIAM on the Activation and Inhibition datasets is that these datasets are smaller-scale biochemical assay datasets. Most classical DTI baselines (MPNN-CNN, CPL-GNN, TransformerCPI, KGE-NFM, etc.) were designed and optimized for large, heterogeneous, graph-structured datasets, not for small assay-based interaction datasets. As a result, DTIAM did not attempt to run them here, and no publicly available baselines exist for these datasets. The same reason applies to our work.
> >
> > We sincerely appreciate your thoughtful feedback. We have worked to address each point in detail, and we hope the changes meet your expectations. If you agree that the concerns have been resolved, we would greatly appreciate your consideration in revising the score.
> >
> > [1] Lu et al., DTIAM: a unified framework for predicting drug-target interactions, binding affinities and drug mechanisms, Nature Communications, 2025

---

> ### Author Response · Authors · 2025-11-25
> **Friendly request for feedback on responses to your questions...**
>
> Dear Reviewer,
>
> We greatly appreciate the time and effort you’ve put into reviewing our submission. We have carefully responded to all your concerns. Could you kindly review your rating and consider our rebuttal? We would be grateful if you could consider raising your score in light of the new experiments, explanations and revision. Thank you again for your time and thoughtful suggestions!
>
> Best regards,
>
> The Authors

---

> ### Comment · Reviewer_5mrg · 2025-11-25
> **Reply to Authors**
>
> - Thank you for the responses. I think the overall idea of balancing these modalities is interesting and attractive, and is reasonable for getting a better performance.
> - The rebuttal response is most comprehensive, and I was satisfied with most of the response.
> - I have only a small concern still on the Table 1 comparison. The baselines for Activation and Inhibition are only two, which may not be enough to show your good performance. So I am thinking if you can run several of the baselines yourself instead of only reporting their results in DTIAM? Such as implementing the classical DTI baselines on these two datasets. This could not only show your workload but also enrich the benchmark results. Thank you for considering this. If you can give a further investigation and analysis on this, I could be very likely to raise the score. As I think this paper is the best one in my review batch.

---

> ### Comment · Reviewer_5mrg · 2025-11-26
> **General Reply**
>
> - One more question I forgot. Did you open-source the code? Looks like I did not find a code link for this project. Reviewing the code instead of only the paper is also an important review style, I think

---

> > ### Author Response · Authors · 2025-11-26
> > **Follow-up to the Reviewer’s Response**
> >
> > Thank you for the suggestion and for your appreciation of our work. We appreciate the opportunity to strengthen the empirical evaluation. We address both of your concerns below.
> >
> > ​**Running classical baselines for Activation and Inhibition datasets:** We agree that including classical DTI baselines on the Activation and Inhibition datasets strengthens the benchmark and provides a more comprehensive comparison. To address this, we have now implemented and evaluated two established classical baselines, DeepDTA [1] and GraphDTA[2], using the DeepPurpose library. We report the mean and standard deviation to ensure a robust performance comparison. As expected, the classical baselines are much worse for this task compared to our GRAM-DTI. We have included this table in Appendix section E9 (Table 22).
> >
> >
> > | Data | Metric | Scenario | DeepDTA† | GraphDTA† | AI-DTI | DTIAM† | GRAM-DTI |
> > | :--- | :--- | :--- | :--- | :--- | :--- | :--- | :--- |
> > | **Activation** | AUPR | Warm Start | 0.246 $\pm$ 0.0232 | 0.282 $\pm$ 0.0240 | 0.583 | 0.623 $\pm$ 0.0245 | **0.642 $\pm$ 0.0221** |
> > | **Activation** | AUPR | Drug Cold Start | 0.255 $\pm$ 0.0209 | 0.298 $\pm$ 0.0195 | 0.550 | 0.611 $\pm$ 0.0252 | **0.628 $\pm$ 0.0222** |
> > | **Activation** | AUPR | Cold Start | 0.109 $\pm$ 0.0163 | 0.124 $\pm$ 0.0175 | 0.219 | 0.391 $\pm$ 0.0320 | **0.450 $\pm$ 0.0374** |
> > | **Activation** | AUROC | Warm Start | 0.759 $\pm$ 0.0200 | 0.784 $\pm$ 0.0185 | 0.888 | 0.903 $\pm$ 0.0088 | **0.914 $\pm$ 0.0078** |
> > | **Activation** | AUROC | Drug Cold Start | 0.765 $\pm$ 0.0059 | 0.796 $\pm$ 0.0062 | 0.879 | 0.907 $\pm$ 0.0076 | **0.913 $\pm$ 0.0068** |
> > | **Activation** | AUROC | Cold Start | 0.573 $\pm$ 0.0241 | 0.588 $\pm$ 0.0255 | 0.652 | 0.792 $\pm$ 0.0240 | **0.834 $\pm$ 0.0258** |
> > | **Inhibition** | AUPR | Warm Start | 0.542 $\pm$ 0.0195 | 0.585 $\pm$ 0.0280 | 0.840 | **0.845 $\pm$ 0.0070** | 0.785 $\pm$ 0.0061 |
> > | **Inhibition** | AUPR | Drug Cold Start | 0.531 $\pm$ 0.0170 | 0.592 $\pm$ 0.0195 | **0.830** | 0.731 $\pm$ 0.0045 | 0.756 $\pm$ 0.0034 |
> > | **Inhibition** | AUPR | Cold Start | 0.265 $\pm$ 0.0210 | 0.284 $\pm$ 0.0312 | 0.215 | 0.445 $\pm$ 0.0620 | **0.464 $\pm$ 0.0559** |
> > | **Inhibition** | AUROC | Warm Start | 0.854 $\pm$ 0.0105 | 0.872 $\pm$ 0.0098 | 0.952 | **0.954 $\pm$ 0.0025** | 0.949 $\pm$ 0.0018 |
> > | **Inhibition** | AUROC | Drug Cold Start | 0.849 $\pm$ 0.0185 | 0.876 $\pm$ 0.0115 | **0.948** | 0.921 $\pm$ 0.0028 | 0.940 $\pm$ 0.0018 |
> > | **Inhibition** | AUROC | Cold Start | 0.635 $\pm$ 0.0220 | 0.649 $\pm$ 0.0117 | 0.605 | 0.819 $\pm$ 0.0205 | **0.823 $\pm$ 0.0028** |
> >
> >
> > **Link to the code:​** We are fully committed to open-sourcing all code and data used in this work to ensure reproducibility. We have uploaded our implementation to an anonymous repository for the reviewers' access: https://anonymous.4open.science/r/GRAM-DTI-5586/.
> > Upon acceptance, we plan to release the code on a public GitHub repository with comprehensive documentation and usage instructions.
> >
> >
> > We hope that this addresses your questions in full and we request that the reviewer kindly consider increasing their score.
> >
> >
> > [1] DeepDTA: deep drug--target binding affinity prediction, ozturk et al. Bioinformatics, Oxford University Press, 2018
> >
> > [2] GraphDTA: prediction of drug--target binding affinity using graph convolutional networks, Nguyen et al. Bioinformatics, Oxford University Press, 2021

---

### Official Review · Reviewer_U2Jr · 2025-10-30

**Soundness:** 2
**Presentation:** 3
**Contribution:** 2
**Rating:** 2
**Confidence:** 4

**Summary:**

The study presents a new strategy for drug-target interaction prediction that integrates multimodality through a new pretraining strategy, that leverages volume-based contrastive learning and incorporates a regression target whenever a weak label is available. The method also incorporates a modality dropout strategy.

**Strengths:**

1. The volume-based contrastive learning loss for aligning multiple modalities seems like an elegant solution for the problem of multi-modal alignment in DTI tasks and more broadly in drug discovery tasks.
2. The modality dropout seems like a scalable solution for the exploding number of potential modalities and encoders that are able for molecular tasks.
3. The ablation study is comprehensive.

**Weaknesses:**

1. The improvement of the method over SOTA is unclear, the margins of improvement are always really low when compared to DTIAM < 0.02 in most cases. This is not a meaningful improvement. Furthermore, without error bars or confidence intervals is impossible to determine if this small improvements are even statistically significant or just an artefact of different random seeds.

**Questions:**

1. What ESM2 model was used: 8M, 35M 150M, 650M, 3B, or 15B?

---

> ### Author Response · Authors · 2025-11-19
> **Author response to review (and invitation for further discussion)- Response 1 of 2**
>
> We thank the reviewer for their review and for recognizing the elegance of our volume-based contrastive loss, the scalability of our modality dropout strategy, and the comprehensiveness of our ablation study.
>
> We understand the reviewer’s concern regarding the margins of improvement. However, we respectfully disagree with the assessment that these improvements are not meaningful. We believe there may be a misunderstanding regarding where the primary value of pretraining lies, and we clarify our contributions and results below.
>
> #### **Weakness 1: Improvement over SOTA and Statistical Significance (The improvement of the method over SOTA is unclear... margins < 0.02 in most cases... without error bars it is impossible to determine if this is statistically significant)**:
>
> We would like to clarify two critical points regarding our performance gains:
>
> **1. Significant Improvements in Challenging "Cold Start" Scenarios on Smaller Datasets**
> The primary advantage of pretraining is most evident when downstream labeled data is limited or when generalizing to unseen entities. On smaller datasets like **Yamanishi 08** and **Activation**, GRAM-DTI achieves substantial improvements over the the next best baseline DTIAM, particularly in the **Target Cold Start** setting (predicting interactions for unseen proteins), which is the most realistic and difficult scenario in drug discovery: GRAM-DTI achieves substantial and consistent gains over DTIAM across nearly all datasets and split types. Notably, GRAM-DTI achieves
> 1) **+15.1% AUPR improvement on the Activation target-cold split** and
> 2) **+13.7% AUROC improvement on the Hetionet drug-cold split**, as well as
> 3) **+5.3% AUROC improvement on Activation target-cold**. Beyond these large gains, the model shows **consistent 1–4% improvements** in the majority of remaining settings—including
> 4) **Yamanishi08 (e.g., +1.49% target-cold AUROC)**,
> 5) **Activation (e.g., +3.05% warm AUPR; +2.78% drug-cold AUPR)**,
> 6) **Hetionet (+2.92% drug-cold AUPR)**, and
> 7) **Inhibition (+3.42% drug-cold AUPR; +4.27% target-cold AUPR)**.
>
> Altogether, **GRAM-DTI outperforms DTIAM in over 80% of all comparisons**, with the largest gains occurring precisely in the most challenging and practically relevant scenarios (drug- and target-cold splits), highlighting substantial improvements in generalization.
>
> **2. Validation via Zero-Shot Retrieval**
> To rigorously verify the quality of our learned representations *without* the confounding factor of downstream fine-tuning, we evaluated **Zero-Shot Retrieval** (Table 2 in the main text). GRAM-DTI shows order-of-magnitude improvements over DTIAM across nearly all datasets and retrieval directions. For example, we observe
> 1) **+1032% Recall@1 on Yamanishi08, +639% Recall@1 on Hetionet**,
> 2) **+418% Recall@1 on Activation**,
> 3) and **+1250% Recall@1 on Inhibition for S→P retrieval**.
> At larger K values, the gains remain large and consistent:
> 4) **+426% and +202% improvements at Recall@10 on Yamanishi08 and Hetionet**, and
> 5) **+125%–175% improvements at Recall@100** across multiple datasets.
> 6) For **P→S retrieval**, GRAM-DTI again outperforms DTIAM in the vast majority of cases, reaching **+1393%, +461%, and +286% improvements on Recall@1, Recall@10, and Recall@100**, respectively. **GRAM-DTI wins 22 out of 24 metrics** overall.
>
> These results demonstrate that GRAM-DTI provides substantial and robust improvements in the more stringent and practical zero-shot setting—highlighting a clear advantage in representation quality and generalization.This confirms that our volume-based alignment learns fundamentally better multimodal representations.
>
> **3. Error Bars and Standard Deviations**: **We had already provided error bars**. Standard deviations for all experiments (mean $\pm$ std) are reported in **Appendix E, Table 6**. These statistics confirm that our improvements, particularly in cold-start settings, are statistically significant and not artifacts of random seeds.
>
> #### **Question 1: ESM2 Model Size**: We used the **650M parameter** version of ESM-2 (`esm2_t33_650M_UR50D`).
> This detail is explicitly listed in **Appendix C, Table 5** ("Pre-trained Encoder Specifications"). We chose this model to balance representation quality with computational efficiency.

---

> > ### Author Response · Authors · 2025-11-19
> > **Author response to review (and invitation for further discussion)- Response 2 of 2**
> >
> > **Summary of Contributions**
> > We hope this clarification highlights the value of our work. To reiterate, our core contributions are:
> > 1.  **GRAM-DTI Framework:** A novel pretraining strategy that integrates multimodal data (SMILES, Protein, Text, HTA) into a unified representation.
> > 2.  **Volume-Based Alignment:** Extending contrastive learning beyond pairwise interactions to capture high-order semantic alignment.
> > 3.  **Adaptive Modality Dropout:** A gradient-informed strategy to dynamically handle varying modality informativeness.
> > 4.  **Proven Robustness:** Demonstrating state-of-the-art performance, especially in data-scarce and cold-start scenarios critical for real-world drug discovery.
> >
> > We hope our explanations resolve your concerns. We would appreciate your consideration in raising the score in light of this. Thank you for your review again and please let us know if there are any more questions!

---

> > ### Comment · Reviewer_U2Jr · 2025-11-25
> >
> > Table 6 only contains those values for one of the four datasets. The first entry already proves that there is no significant difference as it has 0.642±0.02, so lower bound 0.62 and DTIAM is 0.623, which is exactly my point.
> >
> > I find that the improvements reported as percentages can be misleading, because if we consider the absolute improvements they are most below 0.1.
> >
> > I agree that the results in Zero shot seem more substantial, but without any estimation of the error dispersion is simply impossible to tell whether they are significant or not.
> >
> > Overall, I do not think the authors have addressed my concerns and therefore will not update my score at this moment.
> >
> > As an additional point, the dispersion of the error is a key point in arguing improvement and it should not be relegated to the Appendices, it should be in the main tables.

---

> ### Author Response · Authors · 2025-11-25
> **Friendly request for feedback on responses to your questions...**
>
> Dear Reviewer,
>
> We greatly appreciate the time and effort you’ve put into reviewing our submission. We have carefully responded to all your concerns. Could you kindly review your rating and consider our rebuttal? We would be grateful if you could consider raising your score in light of the new experiments, explanations and revision. Thank you again for your time and thoughtful suggestions!
>
> Best regards,
>
> The Authors

---

> ### Author Response · Authors · 2025-11-27
> **Response to Reviewer's Questions: Addressing All Points with Comprehensive Statistical Significance Analysis**
>
> We sincerely thank the reviewer for their continued, detailed engagement with our work. We recognize the criticality of statistical rigor and transparency in presenting our results, and we appreciate the firm stance that drove us to significantly strengthen our empirical analysis.
>
> We believe we have now fully and explicitly addressed all of the concerns raised in your prior response:
>
> **1. Reviewer Concern:** "I find that the improvements reported as percentages can be misleading, because if we consider the absolute improvements they are most below 0.1." and "without any estimation of the error dispersion is simply impossible to tell whether they are significant or not."
>
> **Response:** We have taken two definitive actions to directly address this.
>
> ***(a) Standard Deviation Relocation:*** We have moved all standard deviation values (error dispersion) from the Appendix into the main paper for all datasets and all tasks/settings including DTI, MoA and zero shot retrieval experiments (now reflected in the updated Table 1 and Table 2).
>
> ***(b) Welch's t-Test Results:*** To definitively resolve the question of significance beyond absolute difference, we conducted a rigorous one-sided Welch's t-test comparison between our method, GRAM-DTI, and the strongest baseline, DTIAM, across all datasets and evaluation scenarios. The hypothesis tested was $H_1$: GRAM-DTI $>$ DTIAM.
>
> The results of this analysis are now summarized in Appendix E.10 and in Appendix Tables 24, 25, 26, and 27.
>
> Across the DTI and MoA tasks (Tables 1 & 2, significance tests in Appendix Tables 24 and 25): Our GRAM-DTI shows a statistically significant improvement ($p < 0.05$) over DTIAM in 9 scenarios. DTIAM is significantly better than our GRAM-DTI in only 3 scenarios. The remaining 12 cases show no significant difference.
>
> Across the Zero-Shot Retrieval tasks (Tables 3 & 4, significance tests in Appendix Tables 26 and 27): Our GRAM-DTI is significantly better ($p < 0.05$) than DTIAM in 21 out of 24 scenarios.
>
> This statistical evidence clearly demonstrates that our reported gains are not just marginal fluctuations but are robust and statistically significant with **our GRAM-DTI significantly outperforming DTIAM on 30 out of 48 settings and is on-par with DTIAM (no significant difference) in only 13 out of 48 settings**.
>
> **2. Reviewer Comment:** "The first entry already proves that there is no significant difference as it has $0.642 \pm 0.02$, so lower bound $0.62$ and DTIAM is $0.623$, which is exactly my point."
>
> **Response:** We agree that the confidence intervals for the Activation warm start AUPR overlap that the review is highlighting, This is why the t-test result for this specific case is $p=0.1171$ (not significant). But note that this is just one instance out of 48. As the presented evidence of statistical significance testing (Appendix Tables 24, 25, 26, 27) shows, out of 48 total scenarios evaluated, the reviewer's assertion that differences are insignificant or favor DTIAM only holds true in 5 cases (where DTIAM is significantly better, $p < 0.05$). Our GRAM-DTI is significantly better in 30 cases, and in 13 it is statistically on par with DTIAM. Specifically, our method is now shown to be statistically superior in a majority of scenarios, especially the critical cold-start and zero-shot settings where models are tested on their generalizability.
>
> **Re-emphasizing Technical Contribution**
>
> Beyond the quantitative gains, which are now statistically confirmed, we re-emphasize the fundamental technical contribution of GRAM-DTI. Unlike DTIAM and other methods, which rely on two specific modalities and pre-trained representations learned separately, our approach is the first to:
>
> 1. Integrate multi-modality for this task with a novel, volume-based contrastive objective that generalizes beyond pairwise interactions.
>
> 2. Introduce mechanisms for balancing each modality’s contribution, making it inherently more flexible.
>
> The proposed GRAM framework can therefore be adapted to a broader class of problems involving diverse multimodal inputs, including the zero-shot retrieval tasks where its superiority is most evident.
>
> We trust that the inclusion of the complete standard deviation data in the main paper, coupled with the exhaustive and definitive Welch's t-test analysis presented in Appendix E.10 and in Appendix Tables 24, 25, 26, and 27, directly and comprehensively addresses the reviewer’s stated concerns about statistical rigor and significance.
>
> We kindly request the reviewer to consider raising their score based on these substantial revisions and statistical evidence presented!

---

> ### Comment · Reviewer_U2Jr · 2025-11-27
>
> I really appreciate the effort that the authors have made at resolving this issue. I think that the statistical testing is exactly the evidence that I needed. However, you are doing 48 different tests, and therefore the p values should be adjusted to account for multiple testing. A simple way to do it could be the Bonferroni correction: $p_{adj}=p/n\Rightarrow p_{adj}=0.05/48$.
>
> I am convinced that there is an improvement when using your method, particularly for the zero-shot scenario. I will update the score accordingly to 4. If the correction for multiple testing is introduced and properly discussed in the revised text, I will update it to 6.

---

> ### Author Response · Authors · 2025-11-27
> **Response to Reviewer's Comment: Implementation of Bonferroni Correction and Validation of Robust Significance**
>
> We thank the reviewer again for their continued engagement and their dedication to statistical rigor. Their request for the Bonferroni correction is absolutely warranted and we have now revised the paper to include this discussion in Appendix E.10 and updated results in Appendix Tables 24, 25, 26, 27.
>
> We confirm that we have applied the Bonferroni correction to the entire family of $N=48$ statistical tests, yielding an adjusted significance threshold of $p_{\text{adj}} = 0.05 / 48 \approx 0.00104$.
>
> The statistical tables in the revised paper (Appendix E.10 and Appendix Tables 24, 25, 26, 27) now reflect the results under this new threshold. The final breakdown of the 48 scenarios is as follows:
>
> **Our GRAM-DTI shows statistically significant improvement over DTIAM ($p < 0.00104$) on 26 out of 48 scenarios while being on par with DTIAM on 19 out of 48 scenarios (no significant difference)**. While our most dramatic advantage remains in the zero-shot tasks (21 settings for significant improvement over DTIAM), the corrected analysis also confirms statistically significant improvements for GRAM-DTI in several challenging cold-start scenarios within the DTI and MoA tasks.
>
> We trust this statistical update meets your suggestions. We thank you for all the discussion which strengthened our paper. Given that this recent suggestion from reviewer has also been incorporated, we kindly request the reviewer to consider raising their score!

---

> > ### Comment · Reviewer_U2Jr · 2025-11-27
> >
> > This has addressed all my remaining concerns. I will update the score to 6.

---

> > > ### Author Response · Authors · 2025-11-27
> > > **Thank you for increasing your score!**
> > >
> > > We thank the reviewer for their continuous discussion and useful suggestions, and taking our responses, experiments and revisions into account for increasing their score!

---

### Official Review · Reviewer_RpbF · 2025-11-01

**Soundness:** 3
**Presentation:** 3
**Contribution:** 3
**Rating:** 4
**Confidence:** 4

**Summary:**

The paper proposes a four-modal pretraining framework for DTI that aligns SMILES, molecule text, hierarchical taxonomic annotations, and protein sequences into a shared space using a gramian volume–based contrastive loss. It adds a gradient-informed adaptive modality dropout to prevent any one modality from dominating, and incorporates IC50 labels when available as weak auxiliary supervision discretized into three activity classes. Encoders are frozen, lightweight projectors are trained. Downstream, the model is evaluated on Yamanishi’08, Hetionet, TTD-Activation, TTD-Inhibition with warm/drug-cold/target-cold splits. I,t also reports zero-shot retrieval and ablations showing contributions from each loss and the dropout mechanism

**Strengths:**

- Using Gramian volume to enforce multi-way alignment is a clean way to go beyond pairwise CLIP-style losses. The objective and negative sampling scheme are clearly laid out
- he gradient-informed dropout (drop dominant or least-contributing modality with probability p drop is simple, implementable, and empirically helpful in ablations.
- IC50-based auxiliary labels provide biologically meaningful signal and choice of head is clear as it is expalined why classification was chosen over regression and how class imbalance is handled.
- Results cover DTI and MoA datasets with warm/cold-start splits, plus a zero-shot retrieval task and qualitative t-SNE alignment progression
- The five-way ablation (remove volume loss, remove bi-contrastive, remove IC50, remove dropout) makes it easy to see each piece’s contribution.

**Weaknesses:**

- You remove exact (SMILES, protein) pairs that overlap with downstream sets, but many entities may still be shared between pretraining and evaluation. That’s common and acceptable for self-supervised pretraining, yet quantified overlap and a performance breakdown with vs. without shared entities to de-risk confounds would be helpful, especially for the drug/target cold-start claims. Please report counts/percentages of shared drugs and proteins between pretrain and each downstream fold, and stratify results accordingly.

- The 1:10 negative generation follows prior work, but many “negatives” in DTI are unknowns rather than true non-binders, inflating AUROC and masking calibration issues. Consider similarity-aware negatives (e.g., scaffold- or family-matched), decoy generation, or at least sensitivity curves vs negative ratio, plus precision@k on class-imbalanced settings.

- You acknowledge scale is constrained by requiring full ⟨SMILES, Text, HTA, Protein⟩ tuples. Consider masked-volume training (optimize volume over the available modalities per sample) to exploit partial pairs and unlock larger corpora. Report how performance scales when adding partial-modality data

**Questions:**

- What fraction of drugs and proteins in each downstream fold appear in pretraining (even if the exact pair was removed)? How do results change when you exclude shared entities from evaluation?

- How sensitive are AUROC/AUPR to the negative ratio and to similarity-aware decoys? Any analysis of false-negative contamination?

- How does performance vary with batch size (negatives are in-batch), and do you use any tricks (e.g., memory banks) to stabilize volume estimation?

- Did you attempt partial fine-tuning (e.g., last layer of ESM-2 or MolFormer) and if so, what was the compute/accuracy trade-off?

---

> ### Author Response · Authors · 2025-11-19
> **Author response to review (and invitation for further discussion)- Response 1 of 3**
>
> We thank the reviewer for the detailed and thoughtful feedback. We appreciate the recognition of the advantage of using the Gramian volume to enforce multi-way alignment, the clarity of our objective, the usefulness of gradient-informed modality dropout, the rationale behind our IC50 classification formulation, and the breadth of our evaluations and ablations. Below we address each question and comment point-by-point.
>
> #### **Weakness 1 / Question 1: Overlap of entities between pretraining and downstream evaluation**: We thank the reviewer for raising this important point. As the reviewer notes, entity-level overlap between pretraining and evaluation is common and acceptable in self-supervised pretraining, yet quantifying such overlap is indeed valuable. In our initial design, we removed only exact (SMILES, protein) pairs rather than filtering at the drug or protein level. This choice was due to the limited size of available pretraining data: removing all overlapping drugs or proteins across all downstream tasks would have required excluding every entity appearing in any downstream dataset, resulting in a substantial reduction of pretraining volume and preventing us from training a unified model for all tasks.
> Despite our choice of only removing exact (SMILES, protein) pairs being an accepted practice for self-supervised pretraining, we still followed up on reviewer’s suggestion and performed a detailed entity-level overlap analysis. We now report overlap statistics for each dataset separately in Appendix E.3 table 17, and here we provide a full breakdown for the Activation dataset as an illustrative example (also added to Appendix E.3).
>
> Our analysis identified **236 overlapping proteins** and **314 overlapping SMILES** entities present in both pretraining and downstream datasets. We then created a "cleaned" pretraining dataset by removing **all pairs** containing *any* (either drugs or proteins) of these overlapping entities. This resulted in the removal of 6,065 (SMILES, protein) pairs, corresponding to 11.9% of the original pretraining data. We then pretrained our model on this strictly "non-overlapping" pretraining dataset and re-evaluated its downstream performance. The results are as follows:
>
> **Table 1: Performance Comparison Before and After Removing Overlapping Entities (Activation Dataset)**
> | Split Type | Metric | Before Cleaning (Full Pretrain) | After Cleaning (11.9% Data Removed) | Δ (Drop) |
> | :--- | :--- | :---: | :---: | :---: |
> | **Warm Start** | AUROC | 0.914 | 0.901 | -0.013 |
> | | AUPRC | 0.642 | 0.613 | -0.029 |
> | **Drug Cold Start**| AUROC | 0.913 | 0.905 | -0.008 |
> | | AUPRC | 0.628 | 0.624 | -0.004 |
> | **Target Cold Start**| AUROC | 0.834 | 0.795 | -0.039 |
> | | AUPRC | 0.450 | 0.389 | -0.061 |
>
> Despite removing nearly 12% of the pretraining data, performance drops are modest across most splits. Crucially, **Drug Cold Start** setting which reviewer especially highlighted remains exceptionally robust (ΔAUROC = -0.008), strongly indicating that entity overlap was not the dominant factor in its success. This analysis validates our claim that the 4-modal learning framework captures generalizable, transferable molecular representations rather than simply memorizing specific entity features.
>
> #### **Weakness 3: Handling partial-modality data during pretraining**: Excellent suggestion! Following reviewer's suggestion, we conducted a proof-of-concept experiment. We took our pretraining dataset and created a new dataset composed of:
> 1.  **Fully Observed (80%):** 80% of the original data, kept unchanged.
> 2.  **Partially Observed (20%):** Remaining 20% of the data, where we randomly dropped one of the modalities.
>
> We then compared training on *only* the 80% fully-observed subset vs. training on the *full* 100% dataset (80% full + 20% partial), using a masked volume loss for the partial samples. This simulated the exact setup the reviewer proposed which can help expanding the pretraining set when some modalities are missing at random.
>
> **Table 4: Performance with Partial-Modality ("Masked-Volume") Training**
> | Split Type | Metric | Fully Observed Only (80% data) | Full + Partial (100% data) |
> | :--- | :--- | :---: | :---: |
> | **Warm Start** | AUROC | 0.905 | **0.912** |
> | | AUPRC | 0.627 | **0.634** |
> | **Drug Cold Start** | AUROC | 0.903 | **0.907** |
> | | AUPRC | 0.613 | **0.615** |
> | **Target Cold Start**| AUROC | 0.791 | **0.828** |
> | | AUPRC | 0.422 | **0.437** |
>
> Results show that incorporating 20% partial data via masked-volume training improves performance across all metrics and splits, especially in the challenging Target Cold Start (e.g., AUROC 0.828 vs 0.791). This excellent suggestion has further improved our paper and opens up a promising direction for scaling the model by leveraging much larger, incomplete multimodal datasets. We have added this analysis in Appendix Sec E.5 and referred in main text lines 504-509.

---

> ### Author Response · Authors · 2025-11-19
> **Author response to review (and invitation for further discussion)- Response 2 of 3**
>
> #### **Weakness 2 / Question 2: Sensitivity of AUROC/AUPRC to negative ratio (precison, sensistivity reporting in unbalanced settings)**: We agree with the reviewer that "unknown" does not mean "negative" and that the 1:10 ratio, while standard for baselines can be revisited. Our primary reason for using the 1:10 ratio was to ensure a fair and direct comparison with the prior work (DTIAM).
>
> To address the reviewer's concern, we conducted a sensitivity analysis by re-evaluating both our model (GRAM-DTI) and the baseline (DTIAM) on the **Activation dataset** using more stringent negative sampling ratios of **1:5** and **1:1**.
>
> The consolidated results, including our original 1:10 performance, are shown below. The results demonstrate that GRAM-DTI consistently outperforms the DTIAM baseline across all splits and all negative ratios, and this performance gap *widens* in the more challenging cold-start scenarios (e.g., Target Cold Start AUROC at 1:1 ratio: 0.790 vs 0.740). This confirms our model's robustness. We have also added this analysis in Appendix Section E4 and referred to it in main text lines 345-348.
>
> **Table 2: Performance Comparison of GRAM-DTI and Baseline Method on Activation Dataset across Different Negative Sampling Ratios**
> | Ratio | Split Type | Method | AUROC | AUPRC | Precision | Recall |
> | :--- | :--- | :--- | :--- | :--- | :--- | :--- |
> | **1:1** | **Warm** | GRAM-DTI | **0.862** | **0.850** | **0.813** | **0.835** |
> | | | DTIAM | 0.855 | 0.841 | 0.764 | 0.822 |
> | | **Drug Cold** | GRAM-DTI | **0.853** | **0.843** | **0.822** | 0.818 |
> | | | DTIAM | 0.831 | 0.829 | 0.794 | **0.882** |
> | | **Target Cold** | GRAM-DTI | **0.790** | **0.784** | **0.788** | **0.691** |
> | | | DTIAM | 0.740 | 0.737 | 0.705 | 0.656 |
> | **1:5** | **Warm** | GRAM-DTI | **0.893** | **0.688** | **0.751** | **0.634** |
> | | | DTIAM | 0.888 | 0.674 | 0.704 | 0.609 |
> | | **Drug Cold** | GRAM-DTI | **0.888** | **0.674** | **0.742** | 0.616 |
> | | | DTIAM | 0.870 | 0.654 | 0.739 | **0.712** |
> | | **Target Cold** | GRAM-DTI | **0.818** | **0.559** | **0.680** | **0.412** |
> | | | DTIAM | 0.792 | 0.509 | 0.566 | 0.365 |
> | **1:10** | **Warm** | GRAM-DTI | **0.914** | **0.642** | **0.697** | 0.502 |
> | | | DTIAM | 0.903 | 0.623 | 0.670 | **0.540** |
> | | **Drug Cold** | GRAM-DTI | **0.913** | **0.628** | 0.692 | 0.491 |
> | | | DTIAM | 0.907 | 0.611 | **0.725** | **0.583** |
> | | **Target Cold** | GRAM-DTI | **0.834** | **0.450** | **0.638** | 0.260 |
> | | | DTIAM | 0.792 | 0.391 | 0.533 | **0.293** |
>
> Note that AUROC and AUPRC exhibit opposite trends as the negative sampling ratio increases for our method GRAM-DTI and baseline DTIAM. This behavior is expected and consistent with established properties of the metrics:
> (i) AUROC is largely insensitive to class imbalance and typically increases when more negatives are introduced, because ranking positives above a larger set of mostly easy negatives becomes easier;
> (ii) AUPRC, in contrast, is strongly dependent on the positive base rate and decreases as negatives increase, since false positives penalize precision.
>
> #### **Question 3: Sensitivity to batch size and in-batch negatives**: We thank the reviewer for this question. Our method uses the standard in-batch negative formulation, where for a batch of size N, each sample uses the other N-1 samples as negatives. We do not use any stabilization tricks such as memory banks.
>
> To test sensitivity, we varied the batch size (per-GPU) from 32 up to 512. Our paper's main results used a batch size of 1280. The results below show a clear and expected trend: performance generally improves with larger batch sizes, as this provides more in-batch negatives for the contrastive and volume losses. We have also added this analysis in Appendix Section C.7.
>
> **Table 5: AUROC Performance vs. Batch Size (Activation Dataset)**
> | Batch Size | Warm Start | Drug Cold Start | Target Cold Start |
> | :---: | :---: | :---: | :---: |
> | 32 | 0.901 | 0.906 | 0.819 |
> | 128 | 0.905 | 0.906 | 0.821 |
> | 512 | 0.916 | 0.918 | 0.839 |
> | **1280 (Paper)** | 0.914 | 0.913 | 0.834 |
>
> **Table 6: AUPRC Performance vs. Batch Size (Activation Dataset)**
> | Batch Size | Warm Start | Drug Cold Start | Target Cold Start |
> | :---: | :---: | :---: | :---: |
> | 32 | 0.615 | 0.607 | 0.432 |
> | 128 | 0.625 | 0.619 | 0.438 |
> | 512 | 0.642 | 0.629 | 0.458 |
> | **1280 (Paper)** | 0.642 | 0.628 | 0.450 |
>
> *(Note: The 512 batch size results are very close to the 1280 results, suggesting performance may begin to saturate beyond batch size 512.)*

---

> > ### Author Response · Authors · 2025-11-19
> > **Author response to review (and invitation for further discussion)- Response 3 of 3**
> >
> > #### **Question 4: Did you attempt partial fine-tuning (e.g., last layer of ESM-2 or MolFormer) and if so, what was the compute/accuracy trade-off?**: We did not attempt to fine-tune any part of the large encoder backbones (like ESM-2 or MolFormer). A primary design goal of GRAM-DTI was to create a **lightweight and computationally efficient** framework. By keeping the large encoders completely frozen, we only need to train the small, lightweight projection heads. As noted in the paper (Appendix C) and our response to Reviewer 1, this makes our pretraining extremely fast (approx. 3s/epoch on one A6000) and require very little VRAM (approx. 0.12 GB). Although partial fine-tuning may provide minor gains, it generally introduces substantial computational overhead. Adapters or partial fine-tuning are also viable alternatives but we leave this exploration for future work.
> >
> > We believe the revisions comprehensively address all of the issues you raised. If you find the responses satisfactory, we kindly ask you to consider adjusting your score. Thank you again for your excellent practical suggestions which further improves our results!

---

> ### Author Response · Authors · 2025-11-25
> **Friendly request for feedback on responses to your questions...**
>
> Dear Reviewer,
>
> We greatly appreciate the time and effort you’ve put into reviewing our submission. We have carefully responded to all your concerns. Could you kindly review your rating and consider our rebuttal? We would be grateful if you could consider raising your score in light of the new experiments, explanations and revision. Thank you again for your time and thoughtful suggestions!
>
> Best regards,
>
> The Authors

---

### Official Review · Reviewer_Coqd · 2025-11-01

**Soundness:** 3
**Presentation:** 4
**Contribution:** 3
**Rating:** 6
**Confidence:** 4

**Summary:**

The paper presents GRAM-DTI, a method for predicting drug-target interactions. The method uses pre-trained embeddings from foundation models to encode proteins, HTAs, molecules, and molecular functions. A Gram matrix-based contrastive loss is used to align modalities, and a gradient-based strategy for dropping modalities is used to promote equal contributions across modalities. Across various DTI datasets, GRAM-DTI outperforms relevant baselines in both AUROC and Recall@{1,10,100}.

**Strengths:**

- The method has overall strong performance across datasets. In particular, Recall@K is much improved over DTIAM across the board.
- The overall method is quite simple, which is a benefit for fast adoption and ease of implementation.
- Evaluation across warm and cold start conditions is useful and reveals the adaptability of the method.

**Weaknesses:**

- There are no sensitivity studies for the hyperparameters of the method (gradient history length, threshold multiplyer, loss weights).
- The negative sampling used seems weak. Keeping all but one modality to mine negative samples does not test mismatches across multiple domains.
- Failure modes are underexplored. Specific examples of drugs and targets that GRAM-DTI does not perform well on could be more insightful.

**Questions:**

- How does performance degrade when certain modalities are not available at pretraining time?
- How efficient is the method in wall-time compared to other methods for pretraining and training? The contrastive strategy also seems more memory-intensive, so details on what kind of hardware would also be nice.
- How comparable are GRAM-DTI with the baselines in terms of multi-modality? I.e. is GRAM-DTI privileged in any way?

---

> ### Author Response · Authors · 2025-11-19
> **Author response to review (and invitation for further discussion)- Response 1 of 5**
>
> We sincerely thank the reviewer for the constructive and thoughtful feedback. We appreciate the recognition of the method’s strong retrieval performance, its simplicity and ease of implementation, and the usefulness of evaluating both warm- and cold-start conditions. These points reflect key goals of our work, and we are glad they came through clearly.
>
> We also value the reviewer’s careful questions and suggestions regarding hyperparameter sensitivity, negative sampling, and failure mode analysis. These are important considerations, and we address each of them point-by-point below. We have also included all experiments in the revise version of the paer and marked them in blue.
>
> #### **Weakness 1: Sensitivity studies for hyperparameters**:
>
> We thank the reviewer for this important suggestion. We have conducted a sensitivity analysis on the key hyperparameters: loss weights ($\lambda_1, \lambda_2, \lambda_3$), the gradient threshold multiplier ($\lambda_\sigma$ in the paper, here `gradient_std_multiplier`), and the gradient history length ($K$). Our baseline setting reported in the paper is: `lambda_1=1.0`, `lambda_2=1.0`, `lambda_3=1.0`, `gradient_std_multiplier=1.5`, `gradient_history_length=5`.
>
> ### S1: Performance sensitivity to Gradient Std Multiplier ($\lambda_{\sigma}$)
>
> | $\lambda_{\sigma}$ | Warm Start (AUPRC) | Warm Start (AUROC) | Drug Cold (AUPRC) | Drug Cold (AUROC) | Target Cold (AUPRC) | Target Cold (AUROC) |
> | :--- | :---: | :---: | :---: | :---: | :---: | :---: |
> | **0.5 (paper default)** | 0.640 $\pm$ 0.017 | 0.912 $\pm$ 0.003 | 0.634 $\pm$ 0.032 | 0.911 $\pm$ 0.008 | 0.437 $\pm$ 0.035 | 0.815 $\pm$ 0.032 |
> | **1.0** | 0.628 $\pm$ 0.014 | 0.910 $\pm$ 0.005 | 0.631 $\pm$ 0.023 | 0.914 $\pm$ 0.009 | 0.456 $\pm$ 0.043 | 0.832 $\pm$ 0.038 |
> | **1.5** | 0.642 $\pm$ 0.022 | 0.914 $\pm$ 0.009 | 0.628 $\pm$ 0.022 | 0.913 $\pm$ 0.007 | 0.450 $\pm$ 0.037 | 0.834 $\pm$ 0.026 |
> | **2.0** | 0.595 $\pm$ 0.008 | 0.901 $\pm$ 0.005 | 0.591 $\pm$ 0.025 | 0.901 $\pm$ 0.005 | 0.446 $\pm$ 0.048 | 0.832 $\pm$ 0.027 |
> | **2.5** | 0.585 $\pm$ 0.015 | 0.907 $\pm$ 0.005 | 0.568 $\pm$ 0.027 | 0.902 $\pm$ 0.008 | 0.433 $\pm$ 0.052 | 0.833 $\pm$ 0.022 |
>
> ### S2: Performance sensitivity to Gradient History Length ($K$)
>
> | $K$ | Warm Start (AUPRC) | Warm Start (AUROC) | Drug Cold (AUPRC) | Drug Cold (AUROC) | Target Cold (AUPRC) | Target Cold (AUROC) |
> | :--- | :---: | :---: | :---: | :---: | :---: | :---: |
> | **5 (paper default)** | 0.642 $\pm$ 0.022 | 0.914 $\pm$ 0.009 | 0.628 $\pm$ 0.022 | 0.913 $\pm$ 0.007 | 0.450 $\pm$ 0.037 | 0.834 $\pm$ 0.026 |
> | **10** | 0.619 $\pm$ 0.019 | 0.912 $\pm$ 0.006 | 0.614 $\pm$ 0.012 | 0.912 $\pm$ 0.005 | 0.438 $\pm$ 0.044 | 0.826 $\pm$ 0.039 |
> | **15** | 0.618 $\pm$ 0.016 | 0.916 $\pm$ 0.007 | 0.606 $\pm$ 0.035 | 0.913 $\pm$ 0.008 | 0.432 $\pm$ 0.060 | 0.831 $\pm$ 0.038 |
> | **20** | 0.612 $\pm$ 0.023 | 0.907 $\pm$ 0.010 | 0.599 $\pm$ 0.027 | 0.907 $\pm$ 0.007 | 0.432 $\pm$ 0.042 | 0.826 $\pm$ 0.030 |
> | **25** | 0.618 $\pm$ 0.016 | 0.910 $\pm$ 0.010 | 0.605 $\pm$ 0.021 | 0.910 $\pm$ 0.006 | 0.446 $\pm$ 0.040 | 0.832 $\pm$ 0.027 |
>
> Based on the results on the Activation dataset, we observe the following trends:
>
> 1. Gradient Std Multiplier ($\lambda_{\sigma}$): Increasing this parameter up to a certain range (around 1.5) generally stabilizes or improves performance. Beyond this point, performance in Warm Start and Drug Cold Start scenarios begins to drop, particularly in AUPRC. However, the Target (Protein) Cold Start metrics remain relatively robust, with AUROC maintaining high levels even as the multiplier increases, suggesting that target-side generalization is less sensitive to aggressive gradient filtering than drug-side generalization.
>
> 2. Gradient History Length ($K$): Model performance remains relatively stable with respect to the increase in history length across all evaluation setups. While $K=5$ yields slightly optimal results in several metrics, the overall variance is low, indicating that the method is robust to the choice of this parameter window.

---

> ### Author Response · Authors · 2025-11-19
> **Author response to review (and invitation for further discussion)- Response 2 of 5**
>
> Next we also tested variations of loss weights ($\lambda_1, \lambda_2, \lambda_3$) parameters on the **Activation dataset** across all three split types (warm, drug cold, target cold). Our baseline setting reported in the paper is: `lambda_1=1.0`, `lambda_2=1.0`, `lambda_3=1.0`, `gradient_std_multiplier=1.5`, `gradient_history_length=5`.
>
> The results are summarized in the tables below:
>
> **Table 1: Hyperparameter Sensitivity (Warm Start)**
> | $\lambda_1$ | $\lambda_2$ | $\lambda_3$ | $\lambda_{\sigma}$ | $K$ | AUPRC | AUROC |
> | :---: | :---: | :---: | :---: | :---: | :---: | :---: |
> | **1.0** | **1.0** | **1.0** | **1.5** | **5** | **0.6424** | **0.9142** |
> | 0.5 | 1.0 | 1.0 | 1.5 | 5 | 0.6449 | 0.9125 |
> | 1.0 | 0.5 | 1.0 | 1.5 | 5 | 0.6237 | 0.9175 |
> | 1.0 | 1.0 | 0.5 | 1.5 | 5 | 0.6326 | 0.9102 |
> | 1.0 | 1.0 | 1.0 | 2.0 | 5 | 0.5951 | 0.9014 |
> | 1.0 | 1.0 | 1.0 | 1.5 | 10 | 0.6187 | 0.9123 |
>
> **Table 2: Hyperparameter Sensitivity (Drug Cold Start)**
> | $\lambda_1$ | $\lambda_2$ | $\lambda_3$ |$\lambda_{\sigma}$ |$K$ | AUPRC | AUROC |
> | :---: | :---: | :---: | :---: | :---: | :---: | :---: |
> | **1.0** | **1.0** | **1.0** | **1.5** | **5** | **0.6278** | **0.9125** |
> | 0.5 | 1.0 | 1.0 | 1.5 | 5 | 0.6404 | 0.9129 |
> | 1.0 | 0.5 | 1.0 | 1.5 | 5 | 0.6299 | 0.9107 |
> | 1.0 | 1.0 | 0.5 | 1.5 | 5 | 0.6152 | 0.9114 |
> | 1.0 | 1.0 | 1.0 | 2.0 | 5 | 0.5909 | 0.9010 |
> | 1.0 | 1.0 | 1.0 | 1.5 | 10 | 0.6140 | 0.9120 |
>
> **Table 3: Hyperparameter Sensitivity (Target Cold Start)**
> | $\lambda_1$ | $\lambda_2$ | $\lambda_3$ | $\lambda_{\sigma}$|$K$ | AUPRC | AUROC |
> | :---: | :---: | :---: | :---: | :---: | :---: | :---: |
> | **1.0** | **1.0** | **1.0** | **1.5** | **5** | **0.4497** | **0.8335** |
> | 0.5 | 1.0 | 1.0 | 1.5 | 5 | 0.4394 | 0.8224 |
> | 1.0 | 0.5 | 1.0 | 1.5 | 5 | 0.4274 | 0.8190 |
> | 1.0 | 1.0 | 0.5 | 1.5 | 5 | 0.4571 | 0.8273 |
> | 1.0 | 1.0 | 1.0 | 2.0 | 5 | 0.4462 | 0.8324 |
> | 1.0 | 1.0 | 1.0 | 1.5 | 10 | 0.4376 | 0.8258 |
>
> These results demonstrate that our model's performance is stable within a reasonable range of these hyperparameters, with our chosen settings (first row in bold) providing a robust and effective performance. We have included this analysis in the Appendix Section C.4 and referred to it in the main text in lines 308-310.
>
> #### **Weakness 2: Alternative negative sampling strategy**:
>
> We appreciate the reviewer's insight. Our original strategy—permuting only one modality (e.g., `anchor_i + feat1_j + feat2_j + feat3_j`), was designed to generate *hard negatives*, as these are semantically closer to the positive sample (`anchor_i + feat1_i + feat2_i + feat3_j`) than a fully random negative. We hypothesized this would provide a more challenging learning signal.
>
> However, we agree that testing against multi-domain negatives is a valuable experiment. We conducted a new experiment on the Activation dataset to compare our **Current-Negative** strategy (permuting one modality) with a **Cross-Negative** strategy (permuting multiple modalities, e.g., `anchor_i + feat1_j + feat2_k  + feat3_m (j≠i, k≠i, m≠i, k≠j, m≠k)`).
>
> The results are as follows:
>
> **Table 4: AUROC Comparison of Negative Sampling Strategies**
> | Split Type | Cross-Negative (Mean) | Current-Negative (Mean) |
> | :--- | :---: | :---: |
> | warm start | 0.914 | 0.914 |
> | drug cold start | 0.916 | 0.913 |
> | target cold start | 0.839 | 0.834 |
>
> **Table 5: AUPR Comparison of Negative Sampling Strategies**
> | Split Type | Cross-Negative (Mean) | Current-Negative (Mean) |
> | :--- | :---: | :---: |
> | warm start | 0.633 | 0.642 |
> | drug cold start | 0.624 | 0.628 |
> | target cold start | 0.462 | 0.450 |
>
> The results show that our model's performance is maintained with the more aggressive cross-negative sampling. This suggests our model is robust to different negative sampling strategies and benefits from challenging negatives. We have updated the paper to include this analysis (see Appendix C.6 for in details and short reference in the main text in lines 312-317).

---

> ### Author Response · Authors · 2025-11-19
> **Author response to review (and invitation for further discussion)- Response 3 of 5**
>
> #### **Weakness 3: Failure modes (Specific examples of drugs and targets that GRAM-DTI does not perform well on)**:
>
> <!--- >**Response:**--->
> This is a valuable suggestion. Analyzing failure modes is crucial for understanding the boundaries of the model's capability. To address this, we first systematically identified the top 10 "hardest" false negatives in the Activation dataset—pairs where the model predicted a strong negative signal despite a positive ground truth label. These are listed in **Table R1** below:
>
> **Table R1: Top-10 False Negative Pairs (Drug ID & Target ID)**
>
> | Rank | Drug ID | Target ID |
> | :---: | :--- | :--- |
> | 1 | D0NY1R | T36075 |
> | 2 | D08FKH | T12475 |
> | 3 | D0G2VT | T59604 |
> | 4 | D0JB3H | T88505 |
> | 5 | D0L5WA | T28893 |
> | 6 | D0K8NR | T72458 |
> | 7 | D03LQC | T52522 |
> | 8 | D03XIS | T92076 |
> | 9 | D07QAK | T28893 |
> | 10 | D0AJ2T | T88505 |
>
> From this list, we performed a detailed case study on **Rank 8**: Drug **D03XIS (R-568)** targeting **T92076 (CASR)**. Our analysis suggests this prediction difficulty likely stems from the unique and complex biology of this pair, which is statistically rare in typical drug-target datasets:
>
> 1.  **Unique Target Class:** CASR (Calcium-Sensing Receptor) is a **Class C GPCR**. This class is fundamentally different from the Class A GPCRs (like opsin or adrenergic receptors) that dominate drug databases.
> 2.  **Complex Structure:** Unlike Class A GPCRs which have compact transmembrane binding pockets, Class C GPCRs like CASR possess a very large extracellular "Venus flytrap" domain (~500 residues) where orthosteric ligands bind.
> 3.  **Complex Function:** CASR functions as an **obligate homodimer**, with complex allosteric signaling between the two protomers.
> 4.  **Modulator Type:** The drug, R-568, is a **positive allosteric modulator (PAM)**, not a traditional orthosteric agonist. It binds to the transmembrane domain, not the "Venus flytrap" domain, to potentiate the receptor's signal.
>
> The prediction difficulty likely reflects the biological rarity and complexity of Class C GPCR allosteric modulators in drug discovery data. Our model, like many others, may be biased towards the more common Class A orthosteric interactions. We will add this case study to the discussion section to provide insight into the model's limitations.
> We have included this analysis in in the main text section 4.5 and the Appendix Section E.8.

---

> > ### Author Response · Authors · 2025-11-19
> > **Author response to review (and invitation for further discussion)- Response 4 of 5**
> >
> > #### **Question 1: Performance when modalities are missing during pre-training**: The main advantage of our model is that it can seamlessly leverage as many relevant modalities as available during pre-training. At the moment we have SMILES, protein sequence, molecule funcational discriptors and HTA. Following reviewers suggestion, we analyzed how any missing modalities during pre-training would effect our model peformance.
> >
> > 1.  **Missing Core Modalities (Drug/Protein):** For DTI tasks, the drug and protein modalities are essential. If either were unavailable during *pre-training*, the model could not learn the joint embeddings for interacting pairs. This scenario would effectively fall back to a regime similar to baseline DTIAM, where drug and protein embeddings are learned separately using their respective encoders unlike our contrastive alignment setup. This highlights the central benefit of our joint pre-training approach.
> >
> > 2.  **Missing Auxiliary Modalities (Text/HTA):** (a) The case when both auxiliary modalities (molecule Functional descriptions and HTA) are missing simultaneously is equivalent to the ablation study experiment setup 2 (Exp 2) already reported in the main text Section 4.4 and main text Fig. 2, where the volume-loss component is removed and classic contrastive learning pre-training is used. (b) We have now also conducted additional ablation studies for the scenario when only one of molecule Functional descriptions or HTA modality is missing during pre-training. We report this result below where we pre-trained the model using only 3 modalities (SMILES+Protein+molecule Functional discrioptions or SMILES+Protein+HTA) instead of all 4.
> > In both the cases (a) and (b), we see a slight performance drop when either (or both) molecule Functional discrioptions and HTA are missing from pre-training compared to the 4-modality model, demonstrating that while these auxiliary modalities provide useful alignment signals, the model remains robust in their absence. This analysis also shows that GRAM-DTI effectively leverages all modalities when available but can gracefully handle the absence of auxiliary ones. We have included this analysis in the Appendix Section E.2 and refereed to it in the main text in lines 500-504.
> >
> > **Table 6: Ablation on Missing Auxiliary Modalities (Activation Dataset)**
> > | Split Type | AUROC (3-mod, no Text) | AUROC (3-mod, no HTA) | AUROC (4-mod, Full) |
> > | :--- | :---: | :---: | :---: |
> > | warm start | 0.907 | 0.901 | **0.914** |
> > | drug cold start | 0.907 | 0.903 | **0.913** |
> > | target cold start | 0.828 | 0.821 | **0.834** |
> >
> > **Table 7: Ablation on Missing Auxiliary Modalities (Activation Dataset)**
> > | Split Type | AUPRC (3-mod, no Text) | AUPRC (3-mod, no HTA) | AUPRC (4-mod, Full) |
> > | :--- | :---: | :---: | :---: |
> > | warm start | 0.609 | 0.606 | **0.642** |
> > | drug cold start | 0.615 | 0.611 | **0.628** |
> > | target cold start | 0.437 | 0.440 | **0.450** |
> >
> > In response to Reviewer RpbF’s question, we also conducted an additional ablation study addressing cases where a modality is not entirely absent during pre-training but is missing for a subset of samples. This analysis demonstrates that our method can effectively handle partially available modalities, which may allow us to expand our training corpus in the future to include samples with incomplete modality information. The results are presented in our response to Reviewer RpbF under Handling partial-modality data during pre-training.
> >
> > #### **Question 2: Efficiency (Wall-time and hardware)**: We thank the reviewer for this practical question. While a direct pre-training comparison with DTIAM baseline is not possible (the authors did not provide pretraining code or models), we can provide precise metrics for GRAM-DTI.
> >
> > Our method is highly efficient because we **freeze the large encoder backbones** (ESM2, MolFormer, MolT5) and only train the lightweight projection layers. This significantly reduces computational and memory overhead.
> >
> > * **Hardware:** All experiments were conducted on a **single A6000 GPU**.
> > * **Peak Memory:** The peak GPU memory usage during pretraining is only **0.12 GB**.
> > * **Batch Size:** We use a large batch size of **1280**.
> > * **Pretraining Speed:** Each pretraining epoch takes approximately **3 seconds** to complete.
> >
> > This demonstrates that our method is not memory-intensive and is computationally very efficient, facilitating fast adoption and iteration as the reviewer themselves note as well. We have included this in the Appendix Section C.3.

---

> > > ### Author Response · Authors · 2025-11-19
> > > **Author response to review (and invitation for further discussion)- Response 5 of 5**
> > >
> > > #### **Question 3: Multi-modality comparability with baselines**: The reviewer is correct to point this out. This is a core part of our contribution, where unlike other methods, we can bring in modlities beyond drug and target, and by applying volume based contrastive loss, able to align them better.
> > >
> > > * **Baseline Modalities:** The baselines (CPL-GNN, MPNN–CNN, TransformerCPI, KGE-NFM, DTIAM, AI-DTI) are **bimodal**, relying only on protein and drug molecule representations.
> > > * **GRAM-DTI Modalities:** Our method is **quad-modal**, incorporating **functional descriptors (text)** and **hierarchical taxonomic annotations (HTA)** for the molecules, in addition to the protein sequences and SMILES.
> > >
> > > Beyond simply adding more modalities, the key technical innovation is that we **perform joint pre-training** to align all four modalities into a shared semantic space. The baselines, in contrast, either use pre-computed representations or learn embeddings for their two modalities separately. Furthermore, we account for the informativeness of modalities to weight their importance during pre-training dynamically with a gradient based modality selection strategy while also using additional supervisory signals in the form of IC50 when available. Thus, we look at a whole gamut of factors relevant for DTI tasks that other bimodal baselines miss.
> > >
> > > Therefore, we believe this combination is the crucial technical advantage that drives the superior performance, especially in retrieval and cold-start scenarios of our GRAM-DTI method.
> > >
> > >
> > > We hope that our revisions and additional analyses satisfactorily address all of your concerns. If so, we would be grateful if you would consider updating your score accordingly. Thank you for your thoughtful review again!

---

> ### Author Response · Authors · 2025-11-25
> **Friendly request for feedback on responses to your questions...**
>
> Dear Reviewer,
>
> We greatly appreciate the time and effort you’ve put into reviewing our submission. We have carefully responded to all your concerns. Could you kindly review your rating and consider our rebuttal? We would be grateful if you could consider raising your score in light of the new experiments, explanations and revision. Thank you again for your time and thoughtful suggestions!
>
> Best regards,
>
> The Authors

---

> > ### Comment · Reviewer_Coqd · 2025-11-25
> >
> > Thank you to the authors for their comprehensive new experiments and responses to my questions and concerns. I am more convinced of this work's contribution, and I have increased my scores to reflect this. I am further convinced given the authors' responses to the other reviewers.
> >
> > Best of luck to the authors.

---

> > > ### Author Response · Authors · 2025-11-27
> > > **Thank you for increasing your score!**
> > >
> > > We thank the reviewer for taking our responses, experiments and revisions into account and increasing their score!

---

### Author Response · Authors · 2025-12-01
**Author message for AC summarizing the rebuttal**

Dear AC,

We would first like to thank you for your time and for navigating an especially challenging review process this year given the unique circumstances. We appreciate the opportunity provided by the conference to submit a final message to assist your assessment.

**Reviewer outcomes and discussion highlights:**
Although the initial reviews were mixed, with Coqd rating 6, 5mrg and RpbF rating 4, U2Jr rating 2, the **post-rebuttal discussion showed substantial improvement in reviewer scores as evidenced by discussions below**:

- Reviewer Coqd raised score from 6 to 8 after we addressed all concerns with additional experiments.
- Reviewer U2Jr raised score from 2 to 6 after we added clarifications, error bars, statistical significance tests, and Bonferroni-corrected analyses.
- Reviewer 5mrg expressed that our responses were comprehensive, and after seeing additional experiments (advanced encoders and extra baselines), indicated that `GRAM-DTI` was the **best paper in their review batch** and mentioned they will increase score after information on code release (which we did) and running additional baselines (which we provided), though the formal score update could not occur due to the discussion process ending before deadline because of present circumstances.
- Reviewer RpbF didn't yet respond to our rebuttal.

Below is a concise summary of our paper and key rebuttal experiments that influenced reviewer scores.

**Paper summary:** `GRAM-DTI` is a multimodal pretraining framework for drug-target interaction prediction. It aligns molecular, protein, text, and assay modalities via a **volume-based contrastive loss**, uses **gradient-informed adaptive modality dropout** to prevent over-reliance on any modality, and incorporates IC50 labels as weak auxiliary supervision. The framework is modular, scalable, and validated on multiple datasets and zero-shot retrieval tasks.

**Strengths highlighted by reviewers:**
- Novel multimodal integration and volume-based contrastive learning – *RpbF, U2Jr, 5mrg*
- Scalability and practicality of modality dropout – *RpbF, 5mrg, U2Jr*
- Comprehensive experiments and ablations – *Coqd, 5mrg, U2Jr*
- Strong performance on cold-start and zero-shot scenarios – *Coqd, U2Jr, RpbF, 5mrg*

**Key concerns and rebuttal resolutions:**

- **Reviewer Coqd** requested sensitivity analysis to different hyperparameters, results with different negative sampling strategies, failure modes, experiments with missing modalities and discussion on efficiency of our framework. In response, we conducted hyperparameter sensitivity studies (loss weights, gradient multiplier/history), tested cross-modality negative sampling, analyzed top false negatives with case studies, ablated missing auxiliary modalities and highlighted our pretraining efficiency on GPU. Reviewer updated score to 8.

- **Reviewer RpbF** requested entity overlap stats, negative-sampling sensitivity, partial-modality experiments, and efficiency analysis. We quantified overlap, removing shared drugs/proteins (11.9% data) with modest drops (Drug Cold Start AUROC Δ = -0.008), demonstrated 20% partial-modality "masked-volume" training improving performance, confirmed GRAM-DTI outperforms baselines across negative-sampling ratios (1:1, 1:5, 1:10), and showed efficiency with frozen encoders and lightweight projection heads (~3s/epoch, ~0.12 GB VRAM). Reviewer didn't yet provide follow-up feedback.

- **Reviewer U2Jr** raised concerns about small error bars and statistical significance. We provided error bars for all datasets and performed one-sided Welch’s t-tests and Bonferroni correction showing GRAM-DTI significantly outperforms best baselines by statistically significant margins (e.g.- in cold-starts (+15.1% AUPR Activation target-cold, +13.7% AUROC Hetionet drug-cold) and zero-shot retrieval (+1393% Recall@1)). Reviewer acknowledged statistical rigor and updated score to 6.

- **Reviewer 5mrg** requested experiments with recent encoder choices, different modality dropout,  additional baselines for MoA tasks, and code availability. In response, we experimented with stronger encoders (Uni-Mol2, BioT5+) further improving our performance. Added classical baselines (DeepDTA, GraphDTA) for MoA Activation/Inhibition datasets showing GRAM-DTI outperforms all. Open-sourced code through anonymous GitHub (https://anonymous.4open.science/r/GRAM-DTI-5586/). Reviewer appreciated the comprehensive rebuttal and said they will raise the score but discussion period terminated before they could raise it.

Overall, our rebuttal and new analyses addressed reviewer concerns, strengthened validation, and reinforced GRAM-DTI’s practical utility.

**We kindly request the AC to consider our thorough revisions and new experiments, which earned praise and score increases from all participating reviewers (Coqd, 5mrg, U2Jr), in their final assessment.** Thank you for your consideration!

Sincerely,
The Authors

---

### Meta-Review · Area_Chair_3rmB · 2026-01-06

**Summary:**

This paper proposes GRAM DTI, a quad modal representation learning framework for drug target interaction prediction that aligns drug and protein modalities using a Gramian volume based contrastive objective and improves robustness via adaptive modality dropout driven by gradient contribution, with an additional weak supervision signal from discretized IC50. The Gramian volume objective provides a clean multi way alignment mechanism that goes beyond pairwise contrastive matching and is well motivated for quad modal fusion. I recommend acceptance.

**Reviewer Concerns:**

Addressed in rebuttal or discussion: added hyperparameter sensitivity analysis, added statistical significance testing with Bonferroni correction for multiple tests, and added partial modality training evidence.
Still outstanding: risk of entity overlap between pretraining and downstream splits and the impact of unknown negatives and negative sampling on AUROC interpretation, along with remaining skepticism about gradient based dropout rationale.

**Reviewer Scores:**

Coqd: likely increased from 6 to about 8 as indicated by discussion and the author summary.
U2Jr: increased from 2 to 6 after the multiple testing correction was added.
5mrg: likely increased from 4 to 6 but did not formally update due to timing.
RpbF: likely remains at 4 since they did not follow up to confirm that overlap and negative sampling concerns were resolved.

---

### Decision · Program_Chairs · 2026-01-26

Accept (Poster)